

# Spatial Extent of Future Changes in the Hydrologic Cycle Components in Ganga Basin using Ranked CORDEX RCMs

Jatin Anand [1], Manjula Devak [1], Ashvani Kumar Gosain [1], Rakesh Khosa [1] and Chandrika Thulaseedharan Dhanya [1]

[1] Department of Civil Engineering, Indian Institute of Technology Delhi, New Delhi-110016, India

*Correspondence to:* Dr Chandrika Thulaseedharan Dhanya (dhanya@civil.iitd.ac.in)

**Abstract.** The negative impacts of climate change are expected to be felt over wide range of spatial scales, ranging from small basins to large watersheds, which can possibly be detrimental to the services that natural water systems provide to the society. The impact assessment of future climate change on hydrologic response is essential for the decision makers while carrying out management and various adaptation strategies in a changing climate. While, the availability of finer scale projections from regional climate models (RCM) has been a boon to study changing climate conditions, these climate models are subjected to large number of uncertainties, which demands a careful selection of an appropriate climate model, however. In an effort to account for these uncertainties and select suitable climate models, a multi-criteria ranking approach is deployed in this study. Ranking of CORDEX RCMs is done based on its ability to generate hydrologic components of the basin, i.e., runoff simulations using Soil Water Assessment Tool (SWAT) model, by deploying Entropy and PROMETHEE-2 methods. The spatial extent of changes in the different components of hydrologic cycle is examined over the Ganga river basin, using the top three ranked RCMs, for a period from January 2021 to December 2100. It is observed that for monsoon months (June, July, August and September), future annual mean surface runoff will decrease substantially (-50 % to -10%), while the flows for post-monsoon months (October, November and December) are projected to increase (10- 20 %). While, extremes are seen to be increasing during the non-monsoon months, a substantial decrease in medium events is also highlighted. The increase in wet extremes is majorly supplemented by the increased snowmelt runoff during those months. Snowmelt is projected to increase during the months of November to March, with the month of December witnessing 3-4 times increase in the flow. Base flow and recharge are alarmingly decreasing over the basin. Major loss of recharge is expected to occur in central part of the basin. The present study offers a more reliable regional hydrologic impact assessment with quantifications of future dramatic changes in different hydrological sub-system and its mass-transfer, which will help in quantifying the changes in hydrological components in response to climate change changes in the major basin Ganga, and shall provide the water managers with substantive information, required to develop ameliorative strategies.

**Keywords:** Climate change, Uncertainty, Soil Water Assessment Tool (SWAT), CORDEX, General Circulation Model (GCM), Regional Climate Models (RCM), Ganga River.





# 1 Introduction

The recent years have seen massive industrialization and the amplified use of fossil fuels, which have prompted an extraordinary increase in the concentrations of greenhouse gases in the atmosphere. The climate change due to various anthropogenic activities is considered as "extremely likely", thereby causing detrimental effects for both human and natural systems at varied spatial scales (Franczyk and Chang, 2009; IPCC, 2014; Zhang et al., 2012). One of the most important and immediate concerns is the effect of these alterations in water resources systems both spatially and temporally, especially for the regions where available water reserves are already stressed, due to population growth, industrial development and agricultural needs. In recent decades, an indubitable relation between the climate change and water resources together with the potential impacts of climatic change on water resources and hydrology has gained considerable attention in hydrologic research community (Abbaspour et al., 2009; Akhtar et al., 2008; Amadou et al., 2014; Gardner, 2009). The study of potential impacts of future climate change on water resources, as a way to recognize suitable mitigation and adaptation strategies holds utmost importance. There are many observations supporting the evidence of climate change, such as frequent occurrences of climatic extremes, rising sea level and diminishing snow packs (Amadou et al., 2014; Githui et al., 2009; Shamir et al., 2015; Yu et al., 2002). Since the climate and hydrologic cycle is inter linked, variability in climate is expected to alter both the timings and magnitudes of surface runoff. Consequently, climate change has important ramifications on both existing water resources system as well as future water resources planning and management (Abbaspour et al., 2009; Caballero et al., 2007; Fujihara et al., 2008; Xu et al., 2004). Moreover, the imbalance between the water demands and water supply has been increasing of late due to change in climate. The above stated studies confirm that there still exists a huge gap to relate the alterations in climate with water resources to serve the needs of the decision makers.

To investigate the impact of climate change on future runoff generation, the reliable simulations from various General Circulation Models (GCMs), under different scenarios, are used as inputs to hydrological models. These scenarios are generated by considering the different greenhouse gas emissions and modified initial conditions (Abbaspour et al., 2009; Serrat-Capdevila et al., 2007). Despite the skill of GCMs at global scales, the spatial resolution of climate variable, such as precipitation, which is an important component for hydrological modeling of river system, is still too coarse to be used directly as an input to various hydrologic models. (Akhtar et al., 2008; Fujihara et al., 2008; Orlowsky et al., 2008; Serrat-Capdevila et al., 2007; Steele-Dunne et al., 2008; Troin et al., 2015). Moreover, GCM outputs have a serious spatio-temporal bias, which needs to be corrected before feeding to any hydrologic models, as it may lead to unrealistic simulations. In addition, the accuracy of GCMs varies from one climate variable to other. The main sources contributing to the uncertainty of GCM outputs are due to inadequate knowledge of current system, imperfect representation of various physical processes and incapability to produce inter-annual and inter-decadal variability (Arora, 2001; Christensen et al., 2008; Troin et al., 2015). Owing to the coarse resolution of GCMs, downscaling of climate variables becomes necessary. Apart from model uncertainty, scenario uncertainty and internal variability, the downscaling of these GCM scenarios give birth to one more uncertainty i.e. downscaling uncertainty (Abudu et al., 2012; Liang et al., 2008; Troin et al., 2015). Hence, the outputs derived from downscaled regional climate models (RCM) at relatively finer scales, may aid to limit these uncertainties (Shamir et al., 2015; Troin et al., 2015). The finer resolutions coupled with dedicated physics associated with RCMs can improve the physical processes (Paxian et al., 2016; Vanvyve et al., 2008), pertaining to the variability in regional climate, especially when applied over large and complex basins like Ganga





river basin. However, the comparison between RCM simulations and observed data demonstrates that some systematic biases resulting from imperfect adaptation of physical processes; numerical approximations of model's equations and other assumptions; still prevails within the climate variables (Eden et al., 2014; Fujihara et al., 2008). Given the substantial inconsistencies in RCM simulated fields compared to observations, it is often a

prerequisite to process data before it is fed to hydrological models. This prerequisite process referred, as bias correction, is often necessary for a meaningful translation of climate variables, whereby the RCM results featuring the current prevailing climate are corrected to match with the observations (Johnson and Sharma, 2015; Najafi and Moradkhani, 2015; Troin et al., 2015). Frequently adopted methods for bias correction are Quantile-Quantile (Q-Q) mapping for historic scenario and Equidistant Cumulative Distribution Function (EDCDF) matching

method for future scenario (Amadou et al., 2014; Devak et al., 2015; Johnson and Sharma, 2015; Li et al., 2010; Najafi and Moradkhani, 2015; Troin et al., 2015). Since, the simulation of RCMs incorporates various surface information (soil type, vegetation and hydrology), the selection of RCM is highly region dependent. Hence, ranking of RCMs is desirable before coupling with any modeling study (Behzadian et al., 2010; Raju and Kumar, 2014; Singh et al., 2016; Srinivasa Raju et al., 2016). The ability of regional climate model to reproduce the

climate conditions, over the current period, forms the basis for the selection of model. Hence, the ranking of RCM's by comparing observed runoff with the simulated runoff comprises one of the main objectives of this study.

The future changes in various hydrological components such as runoff, groundwater, and evapotranspiration (ET) can be projected through hydrological modeling. Efficient and reliable simulation of the different hydro-

meteorological conditions presiding within a watershed and quantification of the interrelationships between topography, soil layer, vegetation and land use, is a vital step. It is also known that a better way of managing and assessing available water resources is at basin level (Stehr et al., 2008). The belief, that the climate change would have an evident impact on different hydrologic components, is established by the long-term assessment of catchment's water balance (Abbaspour et al., 2009; Serrat-Capdevila et al., 2007). Hence, watershed models are

essential for studying hydrologic processes and their response to different climate changes (Eckhardt and Ulbrich, 2003; Li et al., 2009; Zeng and Cai, 2014; Zhang et al., 2012). Several physically based hydrological models have been recognized and adopted to simulate hydrological processes in a river catchment. Many of the previous studies were either restricted to the examination of particular components of water balance (For e.g., surface runoff, groundwater recharge and evapotranspiration) or were focused to a particular event of a year and seasonal

processes (For e.g., high flows, low flows, extreme events or seasonal undulations) (Caballero et al., 2007; Fu et al., 2007; Li et al., 2009; Scibek and Allen, 2006). An integrated hydrological simulation approach could benefit in understanding the net impact of climate change in a given region (Islam and Gan, 2015; Oguntunde and Abiodun, 2013).

Following the above discussion, we selected river Ganga as our study basin, to implement the objectives. Being

an agrarian–based country, India's economy is highly dependent on proper water resources planning and management of various basins. In particular Ganga river basin, covering 33% of the total geographic area of Indian-sub continent plays a major role in economy. The catchment of Ganga is distributed unequally among several states and has repetitively facing issues such as floods and drought sequences. Areas, especially, residing in downstream parts of the basin suffers with reduced surface runoff in the dry season and substantial increased

flows during the wet season (Akhtar et al., 2008; Bharati et al., 2011; Gardner, 2009; Rees et al., 2004; Whitehead





et al., 2015). Furthermore, the Ganga river basin is highly governed by snowfall accumulation and snowmelt processes within the catchment. Since snowmelt processes provide most of the surface water during non-monsoon season, its monitoring is important in hydrologic simulations. Any disturbance of the hydrologic characteristics in snow-dominated areas in this region could have major impacts on the available water resources. Furthermore,

the Ganga river basin, which is already under huge irrigation, industrial and domestic pressure, together with the probable consequences of climate change, demand a proper insight and understanding on future climate and their effects on available water wealth. Therefore, in Ganga river basin future water resources assessment, minimizing the uncertainties, is important for decision makers and stakeholders for planning and operation of hydrological installations under climate change.

Hence, the present study focuses on the assessment of the hydrological impacts of climate change on water resources in Ganga river basin for the near (2021-2060) and far (2061-2100) future using different regional climate scenarios. The study aims to assess the available water in the basin on account of varying climatic conditions. In this study, to minimize the climate model uncertainty, firstly the RCMs are ranked based on their ability to mimic the hydrology of the basin with the help of a multi-criterion decision making method, viz. PROMETHEE.

Secondly, top three ranked RCMs obtained through PROMTHEE-based ranking approach are used to determine the impact of climate change at sub-basin level and at a monthly time step for the Ganga basin. Subsequently, impact of climate change on various components of water balance of the basin is assessed, which is further used to quantify the changes in water resources in terms of different hydrologic components.

## 2 Study area

Ganga river basin, which serves an area of about 1.08 Mkm$^2$ and covers a stretch of 1200km, finally meets the Bay of Bengal in the east. The basin is situated in the northern part of the Indian sub-continent, traces between the latitudes of 22°30' and 31°30' North and the longitude of 73°30' and 89°00' East (Fig. 1). The catchment area of 0.86 Mkm$^2$ in India, which is nearly 26 percent of total geographic area of the country, is shared among eleven states (Himachal Pradesh, Uttarakhand, Uttar Pradesh, Madhya Pradesh, Chhattisgarh, Bihar, Jharkhand, Punjab,

Haryana, Rajasthan, West Bengal, and the Union Territory of Delhi). The primary sources of water in the river are precipitation, base flow, and snowmelt water from the Himalayas. Average annual precipitation varies between 300 mm to 2000 mm, with the western part of the catchment getting less precipitation in comparison to the eastern part. Most of this rainfall is concentrated in the monsoon months of June to October, creating low surface runoff conditions in the Ganga river basin and its tributaries, thereby causing dry conditions during non-monsoon periods

(November to May). The temperature during winter season ranges from 2°C to 15°C, while that during summer season varies from 25°C to 45°C (Singh, 1994). The impact of climate change on rainfall and temperature patterns is expected to be widely varying across the catchment, which in turn will affect the spatial and temporal distributions of different constituents of water resources.





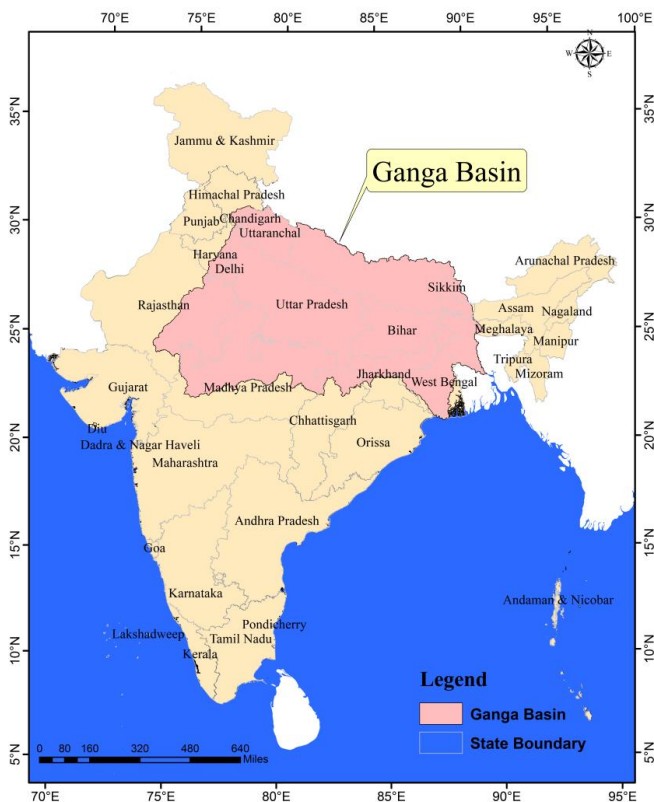

Fig. 1. Location of the Ganga River Basin in India

### 3 Data description

#### 3.1 Observational datasets

5    Daily precipitation and temperature data of 0.5° × 0.5° and 1° × 1° spatial resolution, respectively for the period 1985-2005, provided by India Meteorological Department (IMD) are used as observed datasets. The gridded daily rainfall data is developed by incorporating data from more than 3000 rain gauge stations (quality controlled) using Shepard's interpolation method (Rajeevan et al., 2006). Gridded daily temperature dataset is developed by incorporating temperature gauge data from 395 locations using the same interpolation technique (Srivastava et

10   al., 2009).

Observed daily flows at different gauge stations (Fig. 3) for the period 1990 to 2005, acquired from Central Water Commission (CWC), India are used for the calibration (1990-1999) and validation (2000-2005) of the hydrological model.

#### 3.2 Other hydrological model inputs

15   In addition to the meteorological data (mentioned above), model inputs required to run the hydrological model, include the Digital Elevation Model (DEM), land use layer, soil distribution map of the study area (Table 1).



Table 1. Data description along with its sources used in the SWAT model calibration

| Data | Spatial Resolution | Source |
|---|---|---|
| Digital Elevation Model (DEM) | 90m × 90m | Shuttle Radar Topography Mission (SRTM) (http://www2.jpl.nasa.gov/ srtm/) |
| Soil | 1:5,000,000 scale | Food and Agriculture Organization (FAO) (http://www.fao.org/nr/land/soils/digital-soil-map-of-the-world/en/) |
| Landuse | 1:250,000 scale | National Remote Sensing Center (NRSC) |

**3.3 Climate Model data**

5 In this study, simulations of fourteen RCMs from Coordinated Regional Climate Downscaling Experiment (CORDEX) are extracted (http://cccr.tropmet.res.in/home/ftp_data.jsp). RCMs are chosen based on their data availability over the study region (Table 2). Daily precipitation, maximum and minimum temperature for historical (1985-2005) and future scenario (RCP 4.5, 2021-2100) are extracted from these RCMs.

Table 2. Details of 14 RCMs considered in this study

| S.No. | Institute | RCM | Spatial resolution |
|---|---|---|---|
| 1 | IAES,GUF | CCLM4 | 0.45°×0.45° |
| 2 | CSIRO | CCAM(ACCESS) | 0.5°×0.5° |
| 3 | CSIRO | CCAM(CNRM) | 0.5°×0.5° |
| 4 | CSIRO | CCAM(GFDL) | 0.5°×0.5° |
| 5 | CSIRO | CCAM(MPI) | 0.5°×0.5° |
| 6 | CSIRO | CCAM(BCCR) | 0.5°×0.5° |
| 7 | CCCR, IITM | RegCM4 | 0.45°×0.45° |
| 8 | SMHI | RCA4(CNRM-CERFACS) | 0.45°×0.45° |
| 9 | SMHI | RCA4(NOAA-GFDL) | 0.45°×0.45° |
| 10 | SMHI | RCA4(ICHEC) | 0.45°×0.45° |
| 11 | SMHI | RCA4(IPSL) | 0.45°×0.45° |
| 12 | SMHI | RCA4(MIROC) | 0.45°×0.45° |
| 13 | SMHI | RCA4(MPI-M) | 0.45°×0.45° |
| 14 | GERICS | REMO 2009 | 0.5°×0.5° |





## 4 Model description

### 4.1 The hydrologic simulator (SWAT)

One of the promising hydrological models is SWAT, which has been adjudged by researches as computationally efficient (Abbaspour et al., 2015; Neupane and Kumar, 2015; Shawul et al., 2013). The impacts of climate change on hydrology of the basin are quantified based on the simulations made by SWAT hydrological model developed
by the United States Department of Agriculture (USDA) (Arnold et al., 2013). The SWAT model is a physically based semi-distributed continuous time model that can operate on a large basin and can simulate several processes such as flows in rivers, sediment transport and so on, on a daily/sub-daily time step (Arnold et al., 2013; Neitsch et al., 2002). SWAT simulates various hydrologic processes, including surface runoff generation, using either
SCS curve number method or the Green and Ampt infiltration equation. Model offers vivid methods such as, Hargreaves, Priestley-Taylor or Penman-Monteith methods for the quantification of evapotranspiration. Groundwater flow, lateral flow and percolation are assessed through mass balance of the underlying system. SWAT model assess the impact of land use changes on water supplies and erosion in large-scale catchments. Initially, SWAT simulates each hydrological response unit's (HRU) water balance to estimate the amount of water
available for each sub-basin's main channel at a given step, which is then routed to determine the movement of water through the river system towards the basin outlet.

## 5 Methodology

For the present climate simulation, bias corrected forcings are used as an input for calibrated SWAT model, which is used further for the ranking of the RCMs based on runoff. In the future climate simulations, the top three ranked
RCMs are used to simulate the impact of climate change. Figure 2 illustrates the schematic diagram of the methodology adopted in this study, to achieve the objectives.

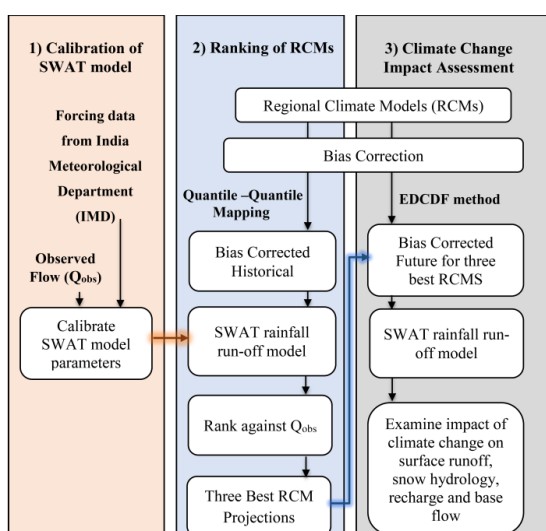

Fig. 2. Schematic diagram of the methodology adopted in the study





### 5.1 Bias correction method

Biases are responsible for poor characterizations of inter annual variability, which is crucial while assessing the impacts of climate change on water resources availability and infrastructure. The quantile-quantile (Q-Q) transformation, also referred to as quantile-quantile mapping or quantile matching, is a widely adopted technique
to bias correct the different climatic variables. Quantile-quantile mapping scales the climate variables according to the observed daily series. This method aims to make the statistical distribution of a climate variable as close as possible to the statistical distribution of the observed variable. Either an empirical distribution or fitted probability distribution is adopted in Q-Q mapping to match the observed and modeled rainfall quantiles (Amadou et al., 2014; Johnson and Sharma, 2015; Li et al., 2010, Troin et al., 2015). In addition to the simplistic application,
quantile-quantile mapping corrects the full distribution of precipitation amounts including the extreme wet and dry events, while maintaining the rank correlation between modeled outputs and observations. This method also takes care of the biases, if any skewness is present in the rainfall distribution. However, an underlying assumption of this method is that the climate distribution does not change much over time i.e., stationarity exists in terms of the variance and skewness of the distribution, and only the mean changes. Hence, Equidistant Cumulative
Distributive Function matching (EDCDFm) method proposed by Li et al. (2010) is deployed in this study, which considers the change of distribution in future. Thus, limitation of Q-Q mapping is dealt by EDCDFm, which adjust the CDF of the model future projections based on the difference between CDFs of model and observed for the considered baseline period. EDCDF method has been widely used in several studies related with climate change and bias correction (e.g., Devak et al., 2015; Wang et al., 2014).

### 5.2 Ranking of RCMs

### 5.2.1 Performance indicators

Different performance indicators are widely used to analyze the performance of the modeled data with respect to the observed time series. Five statistical indicators, namely Nash-Sutcliffe efficiency (NSE), coefficient of determination ($R^2$), normalized root mean square error (NRMSE), absolute normalized mean bias error (ANMBE)
and average absolute relative error (AARE) are considered among the numerous performance indicators available (Moriasi et al., 2007). Nash-Sutcliffe efficiency (NSE) is a normalized statistic that determines the degree of residual variance compared to the measured data variance. NSE indicates how well the computed output matches the observed data along the 45° line (Moriasi et al., 2007). Coefficient of determination ($R^2$) determines the degree of collinearity between measured and computed data. $R^2$ can range from 0 to 1 with higher values indicating good
correlation. Usually higher values of $R^2$ ( >0.5) are considered satisfactory (Moriasi et al., 2007). NRMSE is a measure of the difference between the measured data and the model simulated data. Typically, smaller values of NRMSE suggest better model simulation (Raju and Kumar, 2014). AARE is the average of the absolute values of relative errors. Smaller values indicates better model simulation (Raju and Kumar, 2014). The details of the aforementioned statistical performance measures are tabulated in Table 3.






Table 3. Details of performance measures used in the study

| S.No. | Performance Measures | Formula | Ideal value |
|---|---|---|---|
| 1 | Nash-Sutcliffe efficiency (NSE) | $NSE = 1 - \left[ \dfrac{\sum_{i=1}^{n}\left(X_i - Y_i\right)^2}{\sum_{i=1}^{n}\left(X_i - \overline{X}\right)^2} \right]$ | 1 |
| 2 | Coefficient of Determination ($R^2$) | $R^2 = \dfrac{n\left(\sum_{i=1}^{n} X_i Y_i\right) - \left(\sum_{i=1}^{n} X_i\right)\left(\sum_{i=1}^{n} Y_i\right)}{\sqrt{\left[n\left(\sum_{i=1}^{n} X_i^2\right) - \left(\sum_{i=1}^{n} X_i\right)^2\right]\left[n\left(\sum_{i=1}^{n} Y_i^2\right) - \left(\sum_{i=1}^{n} Y_i\right)^2\right]}}$ | 1 |
| 3 | Normalized Root Mean Square Error ($NRMSE$) | $NRMSE = \dfrac{\left[\dfrac{1}{n}\left(\sum_{i=1}^{n}\left(X_i - Y_i\right)^2\right)\right]^{0.5}}{\overline{X}}$ | 0 |
| 4 | Absolute Normalized Mean Bias Error ($ANMBE$) | $ANMBE = \left|\dfrac{\dfrac{1}{n}\left(\sum_{i=1}^{n}\left(X_i - Y_i\right)\right)}{\overline{X}}\right|$ | 0 |
| 5 | Average Absolute Relative Error ($AARE$) | $AARE = \dfrac{1}{n}\left[\sum_{i=1}^{n}\left|\dfrac{\left(X_i - Y_i\right)}{X_i}\right|\right]$ | 0 |

where, $X$ is the observed data, $Y$ is the simulated data and $n$ is the size of data.

All five statistical indicators are calculated for each of the RCM outputs at all the gauge stations considered in this study. The relative importance of each performance measure is combined by means of weights and finally deriving a common index. Weights are calculated by employing Entropy method and final index for ranking is derived using PROMETHEE-2 method.

### 5.2.2 Entropy method

The entropy method is used to assign weights to the performance indicators. The weights of each of the performance indicators at each of the gauge site depend on the formulated pay off matrix i.e., RCMs versus performance indicator array. The estimation of weights for each of the performance indicator through this approach does not demand any human intervention, which will eliminate any possible bias towards any indicator (Pomerol and Romero, 2000; Raju and Kumar, 2014). Moreover, the variation of weights allows the decision makers to understand the importance of each indicator. The methodology is based on the available information, which is measured by the value of entropy and its relationship with the significance of the criterion. The fundamental idea is that the importance relative to criterion $j$, computed by the weight $w_j$, is a direct function of





the information delivered by the criterion relative to the whole set of alternatives. Greater the dispersion in evaluations of the alternatives, the more important is the criterion. The entropy method works as follows:

    a)   For the given normalized matrix, $p_{ij}$ (where $i$ is the index for RCMs; $j$ is the index for performance indicators), entropy $E_j$ is computed for each of the performance indicator as

$$E_j = -\frac{1}{\ln(N)}\left(\sum_{i=1}^{N} p_{ij} \ln(p_{ij})\right) \qquad (1)$$

where, $i$ is the number of RCMs which varies from $1,2,..., N$ and $j$ is the number of indicators which varies from $1,2,..., J$.

    b)   A higher value of the entropy $E_j$ suggests that the values of normalized matrix $p_{ij}$ are in the close range. However, this is precisely the opposite we require to quantify the segregating power of performance indicator, and thus the opposite value is considered, which is referred to as 'measure of dispersion' or 'degree of diversification' computed as

$$D_j = 1 - E_j \qquad (2)$$

    c)   Normalized weights of indicators are computed as

$$w_j = \frac{D_j}{\sum\limits_{j=1}^{J} D_j} \qquad (3)$$

Higher value of entropy suggests that the uncertainty linked with the performance indicator is very high. Thus, a lower diversification value and subsequently lower value of weight make the performance indicator less important.

### 5.2.3 PROMETHEE-2

The fundamental principle of PROMETHEE-2 is based on a pair-wise evaluation of alternatives along each performance indicator. The method uses the preference function ($P_j(a,b)$) that translates the difference ($d_j(a,b)$) between the performance evaluations ($f_j(a)$ and $f_j(b)$) obtained by two alternatives (i.e. two RCM's ($a$ and $b$) for an indicator $j$), into binary terms (Behzadian et al., 2010; Raju and Kumar, 2014; Srinivasa Raju et al., 2016). Out of the six available criterion functions, usual criterion function, which accounts for even a slight positive difference while comparing two RCMs is implemented in this study. The preference function ($P_j(a,b)$) is given by:

$$P_j(a,b) = \begin{bmatrix} 0 & if & d_j(a,b) \leq 0 \\ 1 & if & d_j(a,b) > 0 \end{bmatrix} \qquad (4)$$

The multicriterion preference index ($\pi(a,b)$) is the weighted average of the preference function ($P_j(a,b)$) for all the indicators and is estimated as

$$\pi(a,b) = \frac{\sum\limits_{j=1}^{J} w_j P_j(a,b)}{\sum\limits_{j=1}^{J} w_j} \qquad (5)$$





The final outranking index ($f^+(a)$), outranked index ($f^-(a)$) and net ranking ($f(a)$) is expressed as

$$\phi^+(a) = \frac{\sum_{i=1}^{N} \pi(a,b)}{(N-1)} \qquad (6)$$

$$\phi^-(a) = \frac{\sum_{i=1}^{N} \pi(b,a)}{(N-1)} \qquad (7)$$

$$\phi(a) = \phi^+(a) - \phi^-(a) \qquad (8)$$

The RCM having the highest $f(a)$ value is considered to be the most suitable RCM and so on. The set of top three RCMs is further utilized to generate future scenarios using ARC SWAT. The runoff and base flow simulations for future scenarios, generated by the SWAT model are then compared with those obtained for the baseline scenario to evaluate the climate change.

## 6 Results and discussion

### 6.1 Development of hydrological model

The ARCGIS based SWAT hydrologic model is calibrated (1990-1999) and validated (2000-2005) for the Ganga river basin. In this study, surface runoff amounts are estimated using the USA Soil Conservation Service, SCS curve number method, while the Penman-Monteith is used for the estimation of potential evapotranspiration. On the basis of DEM, the watershed is divided into 1045 sub-basins, which are further divided into 28648 HRUs that
possess unique landuse/cover, soil and slope. SWAT model is calibrated and validated against observed streamflow at different gauge stations: Farakka (basin outlet), Kuldah Bridge (Tons Basin), Chopan (Son), Triveni (Gandak), Chatara (Kosi), Elgin Bridge (Ghaghara), Dholpur (Chambal), Kalanaur (Yamuna), Rishikesh (Upper Ganga), as shown in Fig. 3. A synoptic representation of the performance of NSE, RSR and PBias at each of the gauge stations considered for the calibration and validation of the basin is shown in Fig. 3. The analysis of
performance measures i.e., NSE, $R^2$, PBias and RSR at various locations depict that performance of the model fall in 'very good' category with NSE>0.70 and $R^2$ >0.80 at most of the gauge stations considered in the study (Fig. 3). High values of NSE and $R^2$ indicate that the model has been able to capture high flows, while the low values of PBias show that the low flows attributed mostly from the snowmelt has also been captured quiet well by the model (Fig. 3). In general, SWAT model is able to capture the intrinsic nature of catchment in terms of
surface runoff very precisely.





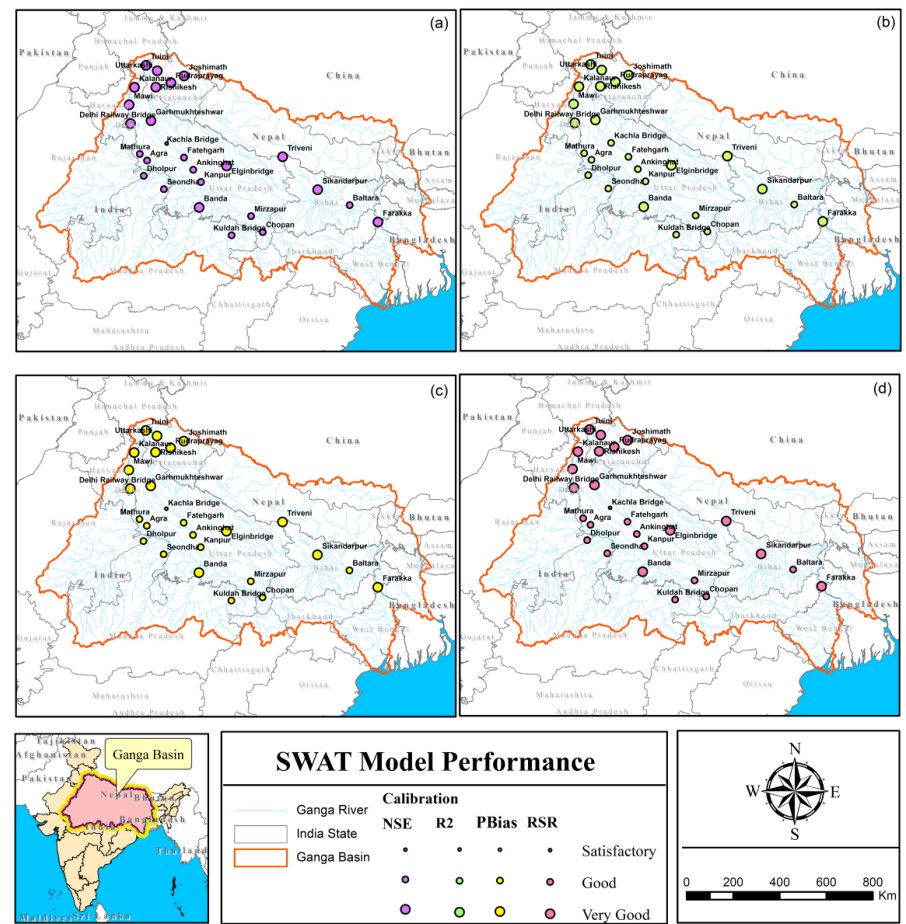

Fig. 3. Values of different performance indicators: (a) NSE. (b) $R^2$, (c) PBias, and (d) RSR indicating the performance of SWAT model in simulating the runoff at different gauge stations in the Ganga river basin.

## 6.2 Ranking of RCMs for Ganga river basin

The calibrated and validated SWAT model for Ganga river basin is used to simulate the surface runoff corresponding to each of the RCMs considered in this study, for the historical period. The surface runoff generated for each of these RCMs are compared with the observed runoff at each of the gauge stations, by computing the performance indicators. For illustration, the performance indicators obtained at Farakka gauge station are shown in Table 4 for the 14 RCMs. From Table 4, it is clear that RCA4 (ICHEC) has performed well at Farakka gauge station with an estimated NSE value of 0.93, whereas a minimum NSE value of 0.19 is computed for RCA4 (IPSL). Similarly, these statistical indicators are computed for the other nine gauge stations. Similar trends are observed in other stations. RCA4 (ICHEC) simulations are relatively better ($R^2$ and NRMSE values of 0.98 and -0.39 respectively) and RCA4 (IPSL) performs badly ($R^2$ and NRMSE values of 0.61 and -1.07 respectively).



Moreover, if ANMBE is considered, RCA4 (CNRM-CERFACS) is the best RCM with value of -0.12, whereas CCAM (Access) is the worst RCM, with an ANMBE value of -0.76. Similarly, for AARE, RCA4 (ICHEC) is the preferred and RegCM4 (LMDZ) is the least preferred, with an estimated value of -0.45 and -2.71 respectively. The above evaluation clearly illustrates that performance indicators behave differently for different RCMs.

For statistical indicators like NSE and $R^2$ value of 1 or close to 1 is desirable whilst for NRMSE, ANMBE and AARE minimum or value of 0 is preferred. Therefore, while the values of NSE and $R^2$ need to be maximized, the values of NRMSE, ANMBE and AARE need to be minimized. Hence, the values of NRMSE, ANMBE and AARE are multiplied by -1 to perform the maximization. The normalized value of every indicator guarantees that the statistical parameters with a relatively larger range do not preside over the indicator with a smaller range. The

total entropy $E_j$ and the normalized weight $w_j$ of each indicator are also given in Table 4 (last row). Of the 5 indicators selected for ranking, ANMBE is the preferred one (with 51% weightage), which implies that its impact on ranking of RCMs is noteworthy on account of its simple but viable deviation principle. The total contribution of NSE, $R^2$ and NRMSE is less than 25%, while AARE contributes 26%.

Table 4. Values of 5-performance measures obtained for the 14 chosen regional climate models (RCMs) for the Ganga river basin at Farakka site, India.

| RCM No. | RCM Name | NSE | $R^2$ | NRMSE | ANMBE | AARE |
|---|---|---|---|---|---|---|
| 1 | CCLM4 (MPI) | 0.65 | 0.89 | -0.98 | -0.70 | -1.77 |
| 2 | CCAM (ACCESS) | 0.70 | 0.94 | -0.96 | -0.76 | -1.68 |
| 3 | CCAM (CNRM) | 0.71 | 0.93 | -0.86 | -0.65 | -1.40 |
| 4 | CCAM (GFDL) | 0.69 | 0.94 | -0.79 | -0.66 | -2.24 |
| 5 | CCAM (MPI) | 0.67 | 0.93 | -0.84 | -0.69 | -2.28 |
| 6 | CCAM (BCCR) | 0.66 | 0.92 | -0.95 | -0.75 | -1.91 |
| 7 | RegCM4 (LMDZ) | 0.81 | 0.92 | -0.51 | -0.21 | -2.71 |
| 8 | RCA4 (CNRM-CERFACS) | 0.81 | 0.91 | -0.55 | -0.12 | -0.67 |
| 9 | RCA4 (NOAA-GFDL) | 0.46 | 0.78 | -0.78 | -0.16 | -1.11 |
| 10 | RCA4 (ICHEC) | 0.93 | 0.98 | -0.39 | -0.24 | -0.45 |
| 11 | RCA4 (IPSL) | 0.19 | 0.61 | -1.07 | -0.16 | -1.45 |
| 12 | RCA4 (MIROC) | 0.87 | 0.94 | -0.46 | -0.15 | -0.70 |
| 13 | RCA4 (MPI-M) | 0.78 | 0.90 | -0.55 | -0.16 | -1.02 |
| 14 | REMO2009(MPI) | 0.47 | 0.77 | -0.96 | -0.49 | -2.36 |
| | **Total Entropy** | **0.98** | **1.00** | **0.98** | **0.92** | **0.96** |
| | **Normalized weights** | **0.11** | **0.02** | **0.11** | **0.51** | **0.26** |

The pairwise preference values (0 or 1) are then calculated for each indicator to estimate the multi-criterion

preference index (Eq. 5). For example, the pair-wise difference between CCLM4 (MPI) and CCAM (ACCESS) for indicator NSE, is -0.05 (0.65-0.70) and hence preference function (Eq. 4) is allotted with the value 0 (since -0.05<0) (Table 5). Diagonal values are assigned 0, on account of the comparison of RCM with itself (Table 5).





Table 5. Multi-criterion preference index values calculated using entropy-based weights for the Ganga River basin at Farakka site for 14 RCMs

| RCM No. | 1 | 2 | … | 7 | 8 | 9 | … | 13 | 14 |
|---|---|---|---|---|---|---|---|---|---|
| 1 | 0.00 | 0.51 | … | 0.26 | 0.00 | 0.13 | … | 0.00 | 0.39 |
| 2 | 0.49 | 0.00 | … | 0.27 | 0.02 | 0.13 | … | 0.02 | 0.39 |
| 3 | 1.00 | 0.98 | … | 0.27 | 0.02 | 0.13 | … | 0.02 | 0.49 |
| 4 | 0.74 | 0.61 | … | 0.27 | 0.02 | 0.13 | … | 0.02 | 0.49 |
| 5 | 0.74 | 0.61 | … | 0.27 | 0.02 | 0.13 | … | 0.02 | 0.49 |
| 6 | 0.24 | 0.61 | … | 0.26 | 0.02 | 0.13 | … | 0.02 | 0.49 |
| 7 | 0.74 | 0.73 | … | 0.00 | 0.12 | 0.24 | … | 0.24 | 0.74 |
| 8 | 1.00 | 0.98 | … | 0.76 | 0.00 | 1.00 | … | 1.00 | 1.00 |
| 9 | 0.87 | 0.87 | … | 0.76 | 0.00 | 0.00 | … | 0.00 | 0.89 |
| 10 | 1.00 | 1.00 | … | 0.49 | 0.49 | 0.49 | … | 0.49 | 1.00 |
| 11 | 0.76 | 0.76 | … | 0.76 | 0.00 | 0.51 | … | 0.51 | 0.76 |
| 12 | 1.00 | 0.98 | … | 1.00 | 0.24 | 1.00 | … | 1.00 | 1.00 |
| 13 | 1.00 | 0.98 | … | 0.76 | 0.00 | 1.00 | … | 0.00 | 1.00 |
| 14 | 0.61 | 0.61 | … | 0.26 | 0.00 | 0.11 | … | 0.00 | 0.00 |

Next step is to calculate the values of outranking index ($\phi$+ (Eq. 6)) and outranked index ($\phi$- (Eq. 7)) by taking the average of preference function row-wise and column-wise, respectively. The calculated value of outranking index ($\phi$+), outranked index ($\phi$-) and net ranking index ($\phi$) (Eqn. 8) corresponding to each RCM are tabulated in Table 6. The corresponding rank of each RCM is also shown in Table 6. RCA4 (CNRM-CERFACS) having the highest value of $\phi$ (0.85) is considered as the best model, followed by RCA4 (MIROC) (rank 2) and RCA4 (ICHEC) (rank 3) with $\phi$ values of 0.80 and 0.53, respectively.

Table 6. Values of $\phi$+, $\phi$–, $\phi$ and ranks of regional climate models (RCMs) for Ganga River basin at Farakka

| RCM No. | $\phi^+$ | $\phi$ | $\phi$ | Rank |
|---|---|---|---|---|
| 1 | 0.21 | 0.79 | -0.57 | 12 |
| 2 | 0.21 | 0.79 | -0.58 | 13 |
| 3 | 0.47 | 0.53 | -0.05 | 8 |
| 4 | 0.34 | 0.66 | -0.32 | 9 |
| 5 | 0.26 | 0.74 | -0.48 | 11 |
| 6 | 0.19 | 0.81 | -0.61 | 14 |
| 7 | 0.50 | 0.49 | 0.00 | 7 |
| 8 | 0.92 | 0.07 | 0.85 | **1** |
| 9 | 0.61 | 0.39 | 0.21 | 5 |
| 10 | 0.77 | 0.23 | 0.53 | **3** |
| 11 | 0.57 | 0.43 | 0.13 | 6 |





| 12 | 0.90 | 0.10 | 0.80 | **2** |
|---|---|---|---|---|
| 13 | 0.74 | 0.26 | 0.49 | 4 |
| 14 | 0.30 | 0.70 | -0.41 | 10 |

The process of ranking of RCMs as discussed above is repeated for all the gauge points considered in this study. Ranking for each system is done independently. Further for whole Ganga basin, ranking is done by considering the outlet at Farakka, which is again an independent study. The ranking of sub-basin is done with a view to provide the researchers and decision makers the set of best RCMs for particular sub-catchments participating in impact studies, if required. Best set of RCMs for all stations is given in Table 7.

Table 7. Ranks of RCMs based on surface runoff estimated from SWAT model for Ganga basin at different locations

| Locations | Rank1 | Rank2 | Rank3 |
|---|---|---|---|
| Farakka | RCA4 (CNRM-CERFACS) | RCA4 (MIROC) | RCA4 (ICHEC) |
| Kuldah Bridge | RCA4 (MPI-M) | RCA4 (MIROC) | RCA4 (CNRM-CERFACS) |
| Chopan | RCA4 (MPI-M) | RCA4 (CNRM-CERFACS) | RCA4 (MIROC) |
| Triveni | REMO2009 (MPI) | CCAM (CNRM) | CCAM (ACCESS) |
| Chatara | REMO2009 (MPI) | CCLM4 (MPI) | CCAM (BCCR) |
| Elgin Bridge | CCAM (CNRM) | CCAM (GFDL) | CCLM4 (MPI) |
| Dholpur | RegCM4 (LMDZ) | REMO2009 (MPI) | CCAM (MPI) |
| Kalanaur | RCA4 (ICHEC) | RCA4 (MIROC) | RCA4 (CNRM-CERFACS) |
| Rishikesh | RCA4 (ICHEC) | RCA4 (CNRM-CERFACS) | RCA4 (MIROC) |
| Kanpur | RCA4 (CNRM-CERFACS) | RCA4 (MIROC) | RCA4 (ICHEC) |

**6.3 Performance analysis of ranked models**

Further, the performances of the ranked RCMs are evaluated by its ability to mimic the hydrology of the catchment. Since, the primary interest lies in the amount of surface runoff generated, emphasis is given on the analysis of surface runoff which is intended to provide indications of the consistency of historical representation of the hydrologic-regime of the catchment when the regional climate models are used as inputs to the hydrological models.

Flow–duration curve is one of the most intuitive way of studying the variability in streamflow through the graphical presentation of distribution of the flow regime. A FDC is constructed by plotting surface runoff against various levels of dependability. Fig. 4 shows flow duration curves (FDCs) for all the ranked RCMs against the observed FDC. The comparison of observed and simulated flows for the top three ranked RCMs demonstrates that in almost all the locations RCM simulations match very well with the observed data. Results also demonstrate the accuracy of RCMs forcings to capture the basic hydrologic properties of the basin in terms of surface runoff




at all the stations except Kuldah Bridge and Dholpur (Fig. 4), where SWAT model over-estimates the runoff values (for all three RCM forcings) in comparison to observed for the same level of dependence or exceedance.

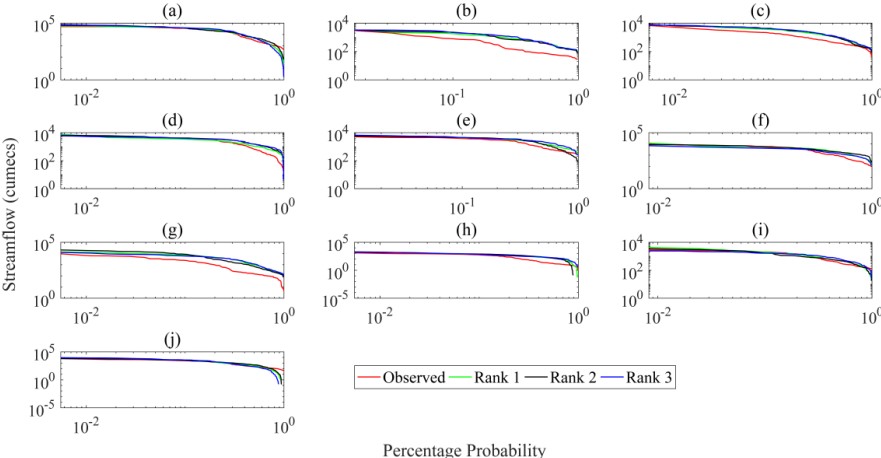

5    Fig. 4. Comparison of Flow Duration Curves (FDC) for (a) Farakka, (b) Kuldah Bridge, (c) Chopan, (d) Triveni, (e) Chatara, (f) Elgin Bridge, (g) Dholpur, (h) Kalanaur, (i) Rishikesh, (j) Kanpur sites

A comparison of long-term average (1990 to 2005) of streamflow for each of the RCMs with the reference streamflow at all the considered stations, as shown in Fig. 5, indicates that the forcings from the top three ranked

10   RCMs are able to capture the long-term average relatively well, at all the stations.

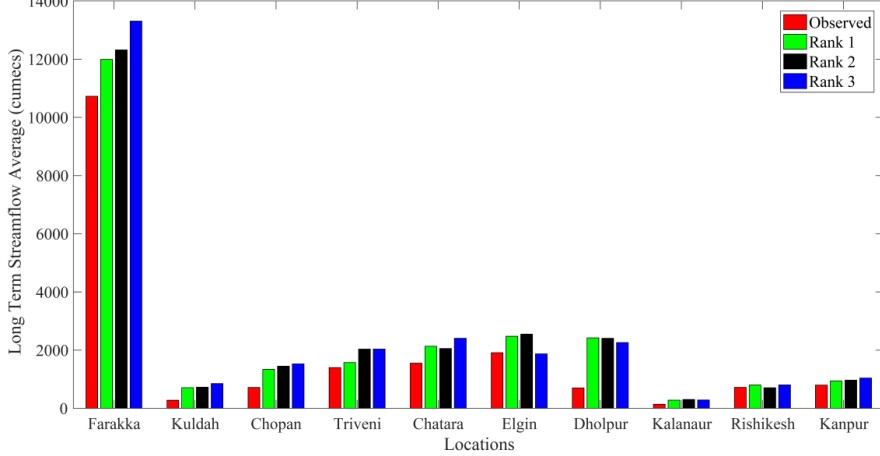

Fig. 5. Comparison of long-term average streamflow at different gauge stations





The performance of RCM models is also compared with respect to their ability to estimate frequency of monsoon and non-monsoon wet and dry flows. Thresholds of 25 percentile and 75 percentile of the observed runoff are fixed to assess the dry and wet events, respectively. Subsequently, the frequencies of these threshold runoff values are then estimated from the SWAT outputs for the top three ranked RCMs. Fig. 6 illustrates the results (a - for
monsoon; and b - for non-monsoon) for the top three ranked RCMs (from the second to the fourth bar) and for the historical period (left most bar) at Farakka gauge station. For the monsoon season, rank 1 and rank 2 models estimated higher frequency of wet and dry monsoon and lower frequency of medium monsoon, whereas ranked 3 model simulated higher frequency of wet and medium and lower frequency of dry monsoon (Fig. 6). All three ranked models gauged higher frequency of wet and medium non-monsoon. Moreover, both rank 1 and rank 3
models over-estimated the dry non-monsoon. The right-most bars in both figures display the average frequencies from three RCMs (Fig. 6).

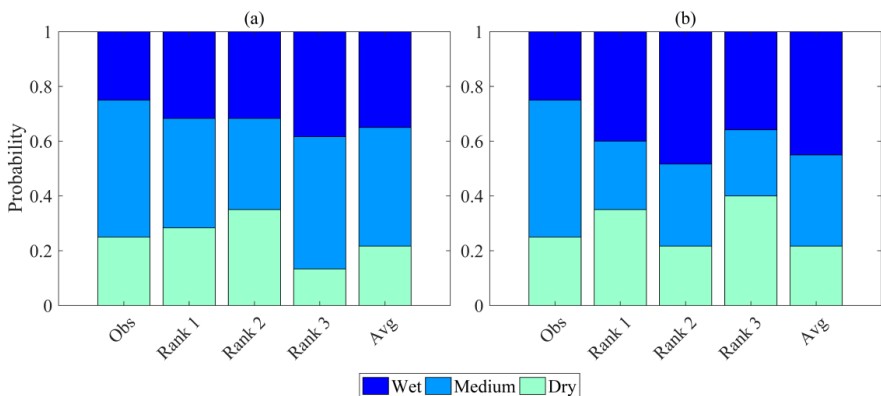

Fig. 6. Frequency of wet, medium, and dry events for both (a) monsoon and (b) non-monsoon months (left and
right panels, respectively) for the period 1990 to 2004. The observed frequency and the average of the three RCMs are also indicated in the left extreme and right extreme bars, respectively.

### 6.4 Future streamflow characteristics in Ganga river basin

The calibrated and validated SWAT model is employed to generate future projections of stream-flows using top
performing RCMs viz. RCA4 (CNRM-CERFACS) (rank 1), RCA4 (MIROC) (rank 2) and RCA4 (ICHEC) (rank 3), at Farakka gauge site, for four 20-year time windows, i.e., mid-century - from 2021 to 2040 and 2041-60 and late-century - from 2061-80 and 2081 to 2100. The future projected alterations in precipitation are reflected by the differences between future and reference surface runoff derived from the simulations of the SWAT model forced by the bias-corrected future projections from the top three ranked RCMs.
The average monthly runoff simulated using the observed climate as well as processed RCMs is presented in Fig. 7. The boxplots indicate the variation of the monthly flows for the 1990-2004 period. The analysis reveals that though the absolute values simulated for three RCMs are relatively different, the overall pattern of the simulated runoff have not changed, when compared with the reference streamflow, the projected peak stream flow occurs in August for both baseline condition and all future periods projected by the RCMs. Fig. 7 depicts the relative





changes in annual monthly streamflow computed by SWAT for the period 2021-40, 2041-60, 2061-80 and 2081-2100. The runoff estimated by SWAT for top performing RCMs suggest that the monthly mean surface runoff estimated by SWAT for monsoon months (June, July, August and September) decreases substantially for the period 2021-40 (~ -50 %), whereas runoff for the months of October, November and December projected a slight

increase (~10 - 30 %) in comparison to the reference mean monthly flows (Fig. 7). For the period 2021-40, RCA4 (CNRM-CERFACS) model projected a maximum decrease in surface runoff of -75%, -60%, -50% in comparison to RCA4 (MIROC) (-50%, -60% and -45%) and RCA4 (ICHEC) (-50%, -51% and -45%) for the months of June, July and August, respectively (Fig. 7). It can be noticed that this trend, similar to 2021-40, continues for 2041-60, 2061-80 and 2081-2100 periods too, wherein all the models projected decrease in surface runoff for the months

of June to August. However, the percentage change reduces from minimum decrease of -30% (RCA4 (ICHEC)), -20% (RCA4 (MIROC)) to -10% (RCA4 (MIROC)) for the month of August, which is the month of peak flow, in 2041-60, 2061-80 and 2081-2100 period, respectively (Fig. 7). Apparently, RCA4 (CNRM-CERFACS) projected maximum decrease and maximum increase in the monsoon months and in the months of October to December, respectively for all considered periods except 2061-80 period for monsoon and 2041-2060 for October

to December. Furthermore, the average monthly stream flows simulated with ranked RCMs during September month, mostly remain in the range of reference flows for all the periods (2021-2100), which implies that exceptional large deviations in the stream flows are improbable during this month. The projected variation in mean flow ranges from -75% (RCA4 (CNRM-CERFACS)) to +140% (RCA4 (MIROC)), -80% (RCA4 (MIROC)) to +60% (RCA4 (MIROC)), -80% (RCA4 (MIROC)) to +60% (RCA4 (MIROC)) and -60% (RCA4

(CNRM-CERFACS)) to +60% (RCA4 (CNRM-CERFACS)) for 2021-40, 2041-60, 2061-80 and 2081-2100 period respectively (Fig. 7). Moreover, for the months of January to April, average monthly runoff projected for the RCMs scenarios by the hydrological model are in the range of reference flows and does not demonstrate any exemplary change (Fig. 7).





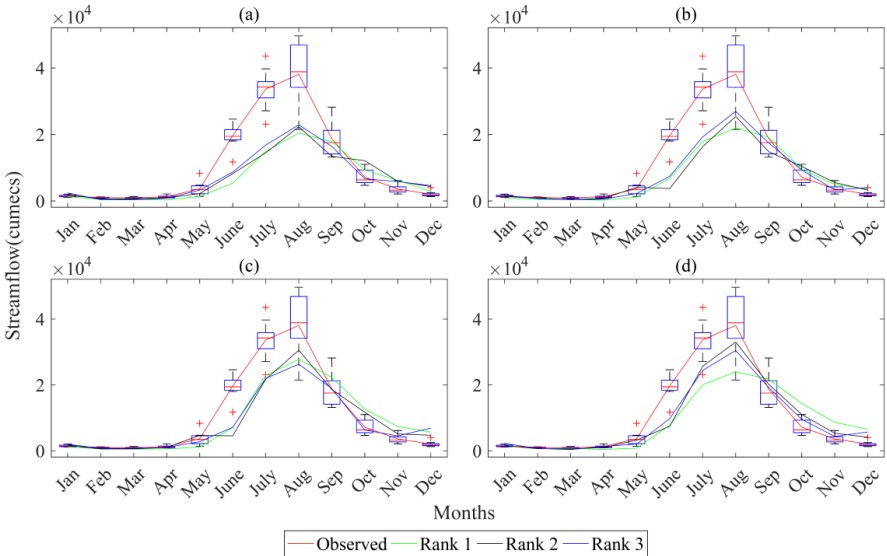

Fig. 7. Average monthly observed and simulated flows for the top ranked RCMs for (a) 2021-40, (b) 2041-60, (c) 2061-80 and (d) 2081-2100 period at Farakka site.

The surface runoff for different decades for monsoon and non-monsoon months, at Farakka site is also demonstrated by means of box plots in Fig. 8. It is noticed that the projections from all RCMs, predicted a decline in the average annual streamflow for Farakka station. Boxplots of average annual streamflow for monsoon season simulated by SWAT (Fig. 8) shows that projected runoff across each decade for all the RCMs decreases with respect to reference flow. However, across the four time windows (2021-2040, 2041-2060, 2061-2080, 2081-2100), model projections estimated a gradual increase in the annual mean streamflow (Fig. 8) from 2021 to the end of the century. In general, the climate scenarios of RCA4 (CNRM-CERFACS) and RCA4 (MIROC) prompt the maximum and the minimum projected decrease, respectively, in the mean annual streamflow during monsoon season in the 21st century for Farakka station of Ganga river basin (Fig. 8). Furthermore, the mean annual streamflow during the non-monsoon season remains practically same for all the decades for all the climate change scenarios (Fig. 8). A significant increase in the extremes is notable for all RCMs in all four periods during the non-monsoon periods, however, which indicates a possible shift in the flow patterns.





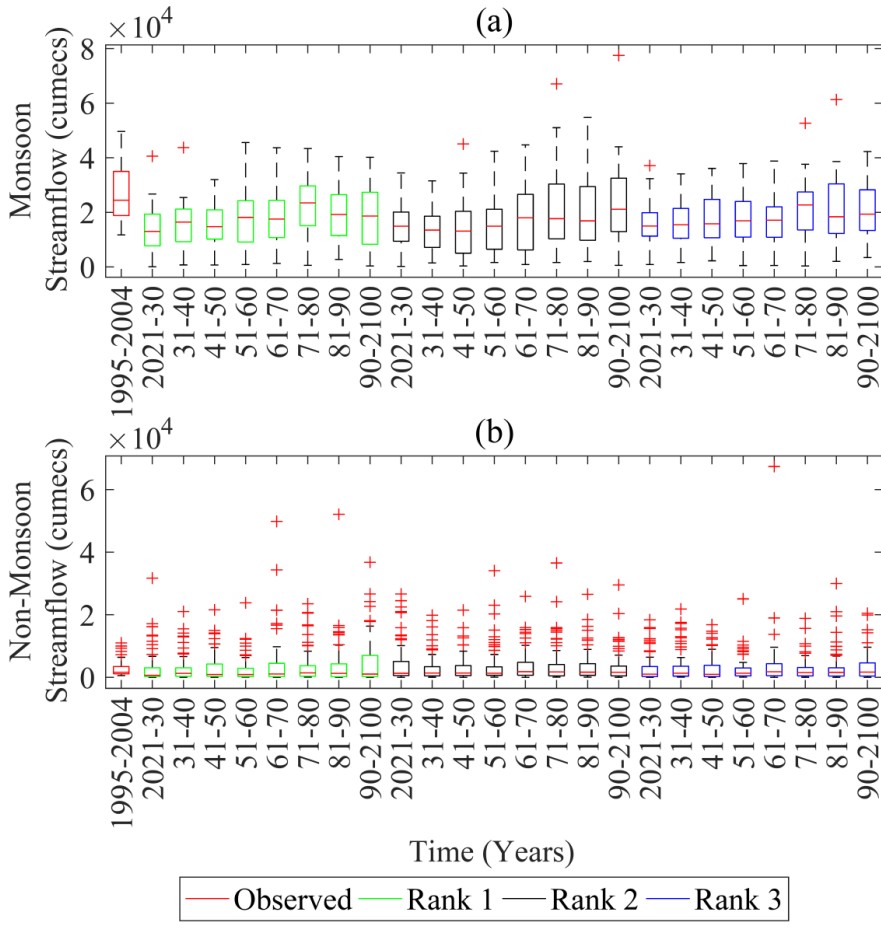

Fig. 8. Box plots of monthly stream-flows in different decades by three RCMs for scenario RCP 4.5 at Farakka site for (a) monsoon months & (b) non-monsoon months.

5    In view of the above inferences, we found it imperative to determine up to what extent the deviations shown in plots such as Fig. 7 and Fig. 8, can be related to individual tributaries of the Ganga river basin. Such an exercise might not only illuminate the impact of climate change on individual tributaries but also provides the degree of severity for individual system. Therefore, the percentage changes in surface runoff with respect to base period (January 1990 to December 2005) are plotted for each of the considered sites lying in different tributaries of

10   Ganga river basin (Fig. 9). As can be noticed from the Fig. 9 that though some of the locations viz. Rishikesh (Upper Ganga) and Kanpur have witnessed increase in the streamflow, Farakka has experienced decrease in the stream flow.





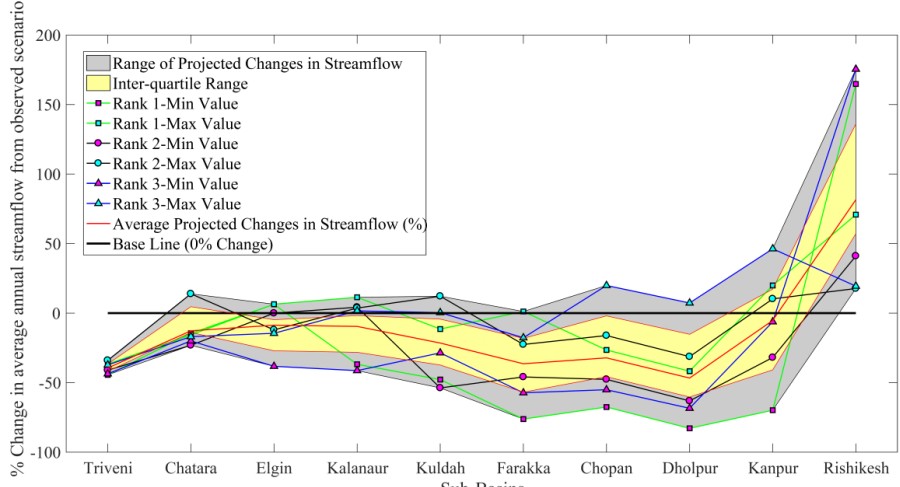

Fig. 9. Range of projected changes in average annual stream flows for different sites of Ganga River basin, Sites are arranged in ascending order of the projected ranges of percentage change. Scatter points (e.g., □, o, Δ) show percentage change as projected by top three RCMs at each location.

In the present study, the SWAT hydrologic model is also used to study the impact of the climate change on the snowfall and snowmelt hydrology. Snow accumulation and snowmelt processes are dominant process in Rishikesh, where snowmelt driven flow provides most of the surface water during non-monsoon season. Fig. 10 shows boxplots of mean annual average streamflow simulated by hydrological model for different climate change

10 scenario and for reference flow for Rishikesh station. The impact of the simulated climate change on the snow cover is found to be very intense, as an increase in surface runoff during the non-monsoon season (Fig. 10 (b)) for the region can be noticed that is attributed mostly to the increased snowmelt within the region. The percentage change of snowmelt in non-monsoon season for the period 20121-2100 is illustrated in Fig. S1 (in Supplementary Material). Months of December and January are witnessed with 3-4 times increase in snowmelt runoff.



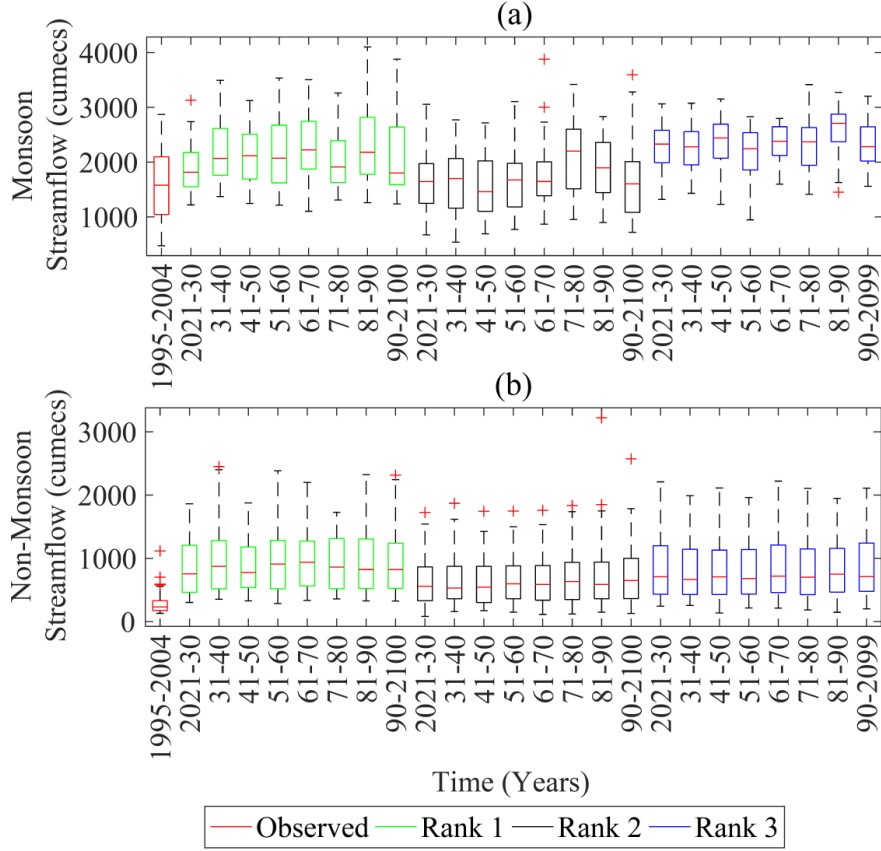

Fig. 10. Box plots of stream-flows in different decades by three RCMs at Rishikesh site for (a) monsoon months & (b) non-monsoon months.

5     Further, changes in wetness indices, on 20-year time-scale, are analyzed. The reference surface runoff can be grouped into three wetness classes for both monsoon and non-monsoon season to study in detail the probability of occurrences of dry (≤ 25 percentile), medium (> 25 percentile and ≤ 75 percentile) and wet flows (> 75 percentile). Projections from three RCMs are compared for monsoon and non-monsoon periods separately, and for all wetness classes with the observed runoff for four 20-year time windows (Fig. 11).





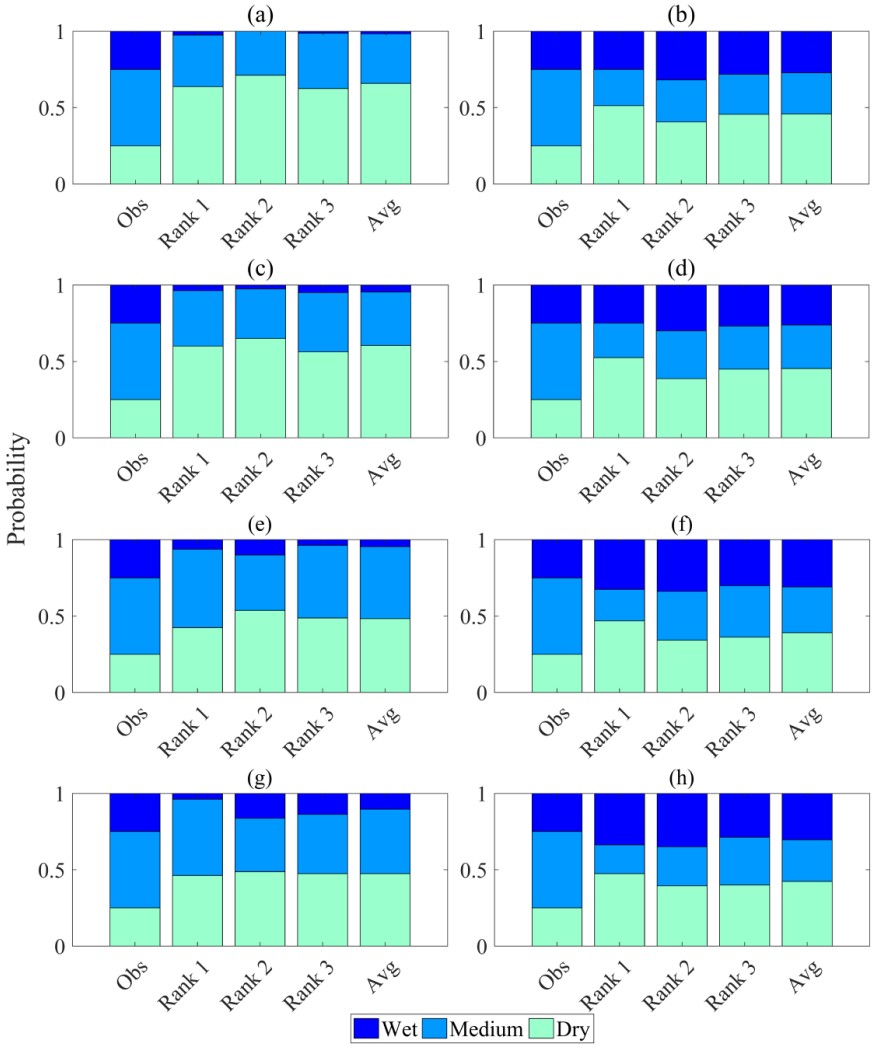

Fig. 11. The projected frequency of wet, medium and dry monsoon months by SWAT model for monsoon (left panel) and non-monsoon months (right panels) for the period from 2021 to 2040 (a & b), 2041 to 2060 (c & d), 2061 to 2081 (e & f) and 2081-2100 (g & h) period by top three ranked RCMs at Farakka outlet.

Fig. 11 compares the results (left panel for monsoon and right panel for non-monsoon) for three climate change projections along with the historical period (left bar). For the monsoon season, all the three RCMs projected higher frequency of dry monsoon and comparatively lower frequency of wet monsoon for 2021-40 period. Similar to the period 2021-40, for rest of the periods viz. 2041-60, 2061-80 and 2081-2100 period, all the three RCMs also

10    projected higher frequency of dry monsoon and lower frequency of wet monsoon. Furthermore, only RCA4 (MIROC) and RCA4 (ICHEC) projected a closer frequency of wet and dry monsoon to base-period in 2081-2100 period (Fig. 11).





It is also evident from Fig. 11 that all three-climate change scenarios projected a reduction in the medium events during non-monsoon periods, resulting into higher frequency of dry and wet events, which in turn indicates an increase in the extremes (relative) during the non-monsoon period. The projections from the average of three RCMs also highlight a similar outcome (Fig. 11).

Another important information requirement for water resources management is that of base flow and recharge situation within each of the sub-basins and subsequently within the basin. SWAT simulated base flow and recharge for the top performing RCMs is compared for all the sub-basins with simulated flow for the base period. As an illustration, Fig. 12 shows the difference between base flow simulation for the RCA4 (CNRM-CERFACS) for the nearby future (2021-60) and distant future (2061-00) with base period (1990-2004), which range from

+5000 to –1000 mm (+ 100% to -500%). A positive difference signifies that base flow simulated for forcing data for future is higher than that of base period and vice-versa. Right bank rivers (Chambal, Sindh and Son (Fig. 12)), part of Haryana and Uttar Pradesh are the regions where base flows for future has been higher than the base period, whereas considerable portion of the basin has showed decrease in the base flows during 2021-60 periods (Fig. 12). The examination of Fig. 12 suggests that the difference between the base period and future reduces for the

negative change and increases for positive change for the 2061-00 period as compared to 2021-60 period. Fig. 12 also shows the difference in recharge values for the RCA4 (CNRM-CERFACS) for the nearby future (2021-60) and distant future (2061-00) with base period (1990-04) which range from +2000 to -1000 mm (+97% to -600%). It can be noticed that the central part of the basin has undergone major loss in the recharge. Similar to the trend in base flow, recharge has also seen a positive increase in 2061-2100 from 2021-60 period (Fig. 12). Similar trends

in base flow and recharge are recorded for RCA4 (MIROC) and RCA4 (ICHEC) (Fig. S2 and Fig. S3 in Supplementary Material).



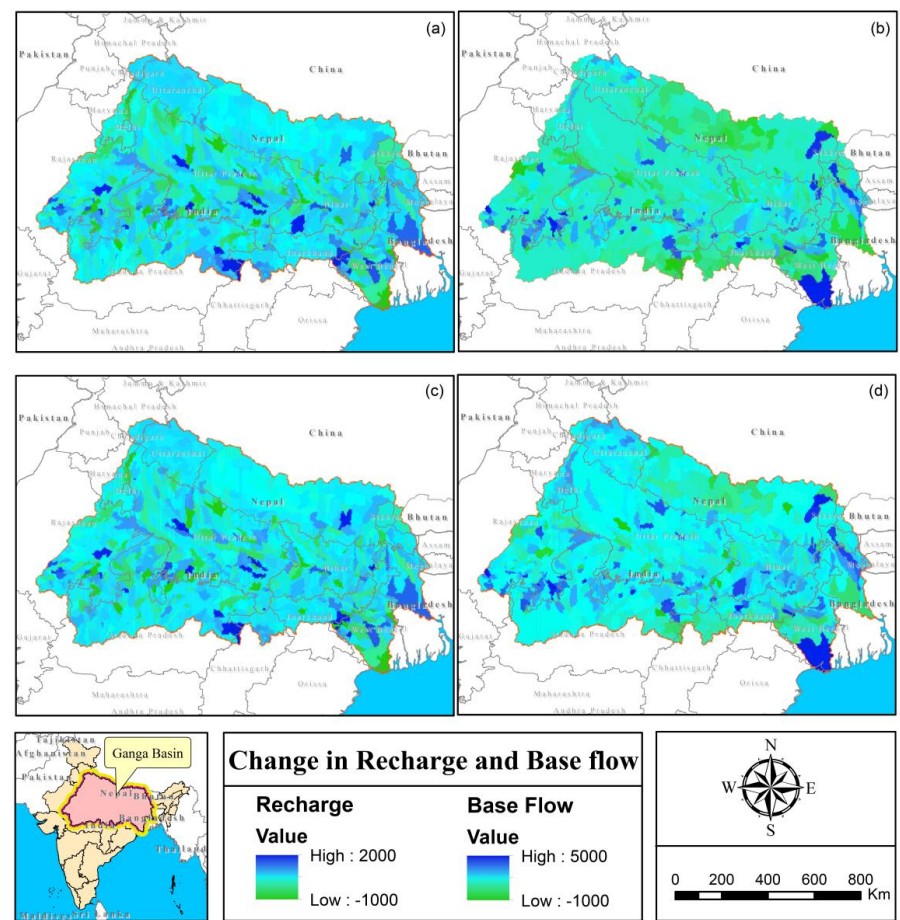

Fig. 12. Change in Recharge for the period (a) 2021-60 and (c) 2061-2100; and Base flow for the period (b) 2021-60 and (d) 2061-2100 in Ganga river basin projected by RCA4 (CNRM-CERFACS)

5      **7 Conclusions**

The prediction of impact of climate change on the hydrological cycle is a very intricate task. At present, the climate change impact study is of type "IF…. THEN", i.e., the consequence of climate change on surface runoff, which differs regionally among different climate change scenarios. The water wealth available for the study region is already under pressure due to ever increasing population and other demands. This study is vital on the ground

10     that it would improve the comprehension of the effect of different climatic change situations on surface runoff, which can be utilized for better administration and advancement of water assets in the concerned area.

The analysis of projected hydrological changes in the region due to climate change is done in a systematic manner. First of all 5 statistical indicators (NSE, $R^2$, NRMSE, ANMBE and AARE) were used to rank 14 RCMs for Ganga River basin using pre-calibrated SWAT model for surface runoff. Relative importance in terms of normalized





weights of the statistical indicators was done using the Entropy method and was further used by PROMETHEE-2 method to derive the final index, which was finally used to rank the 14 RCMs. The study gave a chance to evaluate the aptness of the PROMETHEE-2 methods, which is the first time application of multi-criterion decision making technique to a calibrated and validated SWAT model output viz. surface runoff in the Ganga basin to our

knowledge, along with the Entropy method. RCA4 (CNRM-CERFACS), RCA4 (MIROC) and RCA4 (ICHEC) occupied the first, second and third positions, respectively, for the Farakka gauge site. It may be noted that the resultant ranking patterns may change with the addition of other/more indicators and different RCMs. Successively, outputs of the ranked RCMs were employed to project surface runoff using SWAT. The analysis of projected runoff for the top three ranked models suggest that there is decreasing trend in surface runoff at the

outlet (Farakka) for the monsoon season (June to September) for all the RCMs whereas an increase in the base flow (October to December) has been estimated by all the RCMs. However, the months of January to May show no sudden changes in runoff simulations for all the RCMs. Analysis of projections shows an increase in dry (< 25 percentile) monsoon events, however the probability of wet, dry and medium for non-monsoon remains almost same for all the decades. Further analysis of hydrologic components such as recharge and base flow suggest that

most of the areas have undergone decrease in recharge and base flow. Analysis of snowmelt hydrology has indicated that the snow-melt has increased during the months of November to March with a maximum increase in the month of December. These, spatio-temporal variations projected will provide important insights of surface runoff to climate change projections and therefore, will help in better administration and management of available resources.

Since, SWAT hydrologic model was setup for 1045 sub-basins for the Ganga river basin taking outputs at daily as well as monthly time steps, a substantial amount of information can be obtained for each of the sub-basin. In this study only an outline of the impact of climate change on the different hydrologic components viz. surface runoff, base flow and recharge for the entire basin is provided. However, each of the sub-basins are influenced differently and these analyses can be used to formulate different strategies to reduce the impact of climate change.

Furthermore, the simulation results shown in this study are only acceptable under current land use scenarios in the study region. Considering this, it is necessary to integrate land cover change that is expected in the coming decades and thus associated impacts should be assessed and should be planned for by adopting different irrigation and management practices. Moreover, the present study did not incorporate future water quality aspects of the Ganga river basin but the present model framework can act as the starting point for the future research.

**8 Data availability**

Observed temperature and precipitation data are available with Indian Meteorological Department (IMD) and are available for a reasonable price. The custodian of the surface runoff data used for the calibration and comparison purpose is Central Water Commission (CWC), Ministry of Water Resources, India. All other data sources/links are indicated in section 3.


*Competing interests.* The authors declare that they have no conflict of interest.

*Acknowledgements.* The authors would like to thank Indian Institute of Technology Delhi (India) for providing support for conducting this study.


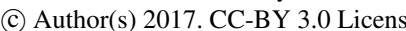


*Author contribution.* JA, MD and CTD designed the experiments. JA and MD developed the model and performed the simulations. JA, AKG and CTD analyzed the results and all authors contributed to the manuscript writing.

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





Table1. Data description along with its sources used in the SWAT model calibration

| Data | Spatial Resolution | Source |
|---|---|---|
| Digital Elevation Model (DEM) | 90m × 90m | Shuttle Radar Topography Mission (SRTM) (http://www2.jpl.nasa.gov/ srtm/) |
| Soil | 1:5,000,000 scale | Food and Agriculture Organization (FAO) (http://www.fao.org/nr/land/soils/digital-soil-map-of-the-world/en/) |
| Landuse | 1:250,000 scale | National Remote Sensing Center (NRSC) |

5    Table 2. Details of 14 RCMs considered in this study

| S.No. | Institute | RCM | Spatial resolution |
|---|---|---|---|
| 1 | IAES,GUF | CCLM4 | 0.45°×0.45° |
| 2 | CSIRO | CCAM(ACCESS) | 0.5°×0.5° |
| 3 | CSIRO | CCAM(CNRM) | 0.5°×0.5° |
| 4 | CSIRO | CCAM(GFDL) | 0.5°×0.5° |
| 5 | CSIRO | CCAM(MPI) | 0.5°×0.5° |
| 6 | CSIRO | CCAM(BCCR) | 0.5°×0.5° |
| 7 | CCCR, IITM | RegCM4 | 0.45°×0.45° |
| 8 | SMHI | RCA4(CNRM-CERFACS) | 0.45°×0.45° |
| 9 | SMHI | RCA4(NOAA-GFDL) | 0.45°×0.45° |
| 10 | SMHI | RCA4(ICHEC) | 0.45°×0.45° |
| 11 | SMHI | RCA4(IPSL) | 0.45°×0.45° |
| 12 | SMHI | RCA4(MIROC) | 0.45°×0.45° |
| 13 | SMHI | RCA4(MPI-M) | 0.45°×0.45° |
| 14 | GERICS | REMO 2009 | 0.5°×0.5° |



Table 3. Details of performance measures used in the study

| S.No. | Performance Measures | Formula | Ideal value |
|---|---|---|---|
| 1 | Nash-Sutcliffe efficiency (NSE) | $NSE = 1 - \left[ \dfrac{\sum\limits_{i=1}^{n}\left(X_i - Y_i\right)^2}{\sum\limits_{i=1}^{n}\left(X_i - \overline{X}\right)^2} \right]$ | 1 |
| 2 | Coefficient of Determination ($R^2$) | $R^2 = \dfrac{n\left(\sum\limits_{i=1}^{n} X_i Y_i\right) - \left(\sum\limits_{i=1}^{n} X_i\right)\left(\sum\limits_{i=1}^{n} Y_i\right)}{\sqrt{\left[ n\left(\sum\limits_{i=1}^{n} X_i^2\right) - \left(\sum\limits_{i=1}^{n} X_i\right)^2 \right]\left[ n\left(\sum\limits_{i=1}^{n} Y_i^2\right) - \left(\sum\limits_{i=1}^{n} Y_i\right)^2 \right]}}$ | 1 |
| 3 | Normalized Root Mean Square Error (NRMSE) | $NRMSE = \dfrac{\left[ \dfrac{1}{n}\left( \sum\limits_{i=1}^{n}\left(X_i - Y_i\right)^2 \right) \right]^{0.5}}{\overline{X}}$ | 0 |
| 4 | Absolute Normalized Mean Bias Error (ANMBE) | $ANMBE = \left| \dfrac{\dfrac{1}{n}\left( \sum\limits_{i=1}^{n}\left(X_i - Y_i\right) \right)}{\overline{X}} \right|$ | 0 |
| 5 | Average Absolute Relative Error (AARE) | $AARE = \dfrac{1}{n}\left[ \sum\limits_{i=1}^{n}\left| \dfrac{\left(X_i - Y_i\right)}{X_i} \right| \right]$ | 0 |

where, $X$ is the observed data, $Y$ is the simulated data and $n$ is the size of data.



Table 4. Values of 5-performance measure obtained for the 14 chosen regional climate models (RCMs) for the Ganga river basin at Farakka site, India.

| RCM No. | RCM Name | NSE | $R^2$ | NRMSE | ANMBE | AARE |
|---|---|---|---|---|---|---|
| 1 | CCLM4 (MPI) | 0.65 | 0.89 | -0.98 | -0.70 | -1.77 |
| 2 | CCAM (ACCESS) | 0.70 | 0.94 | -0.96 | -0.76 | -1.68 |
| 3 | CCAM (CNRM) | 0.71 | 0.93 | -0.86 | -0.65 | -1.40 |
| 4 | CCAM (GFDL) | 0.69 | 0.94 | -0.79 | -0.66 | -2.24 |
| 5 | CCAM (MPI) | 0.67 | 0.93 | -0.84 | -0.69 | -2.28 |
| 6 | CCAM (BCCR) | 0.66 | 0.92 | -0.95 | -0.75 | -1.91 |
| 7 | RegCM4 (LMDZ) | 0.81 | 0.92 | -0.51 | -0.21 | -2.71 |
| 8 | RCA4 (CNRM-CERFACS) | 0.81 | 0.91 | -0.55 | -0.12 | -0.67 |
| 9 | RCA4 (NOAA-GFDL) | 0.46 | 0.78 | -0.78 | -0.16 | -1.11 |
| 10 | RCA4 (ICHEC) | 0.93 | 0.98 | -0.39 | -0.24 | -0.45 |
| 11 | RCA4 (IPSL) | 0.19 | 0.61 | -1.07 | -0.16 | -1.45 |
| 12 | RCA4 (MIROC) | 0.87 | 0.94 | -0.46 | -0.15 | -0.70 |
| 13 | RCA4 (MPI-M) | 0.78 | 0.90 | -0.55 | -0.16 | -1.02 |
| 14 | REMO2009(MPI) | 0.47 | 0.77 | -0.96 | -0.49 | -2.36 |
| | **Total Entropy** | **0.98** | **1.00** | **0.98** | **0.92** | **0.96** |
| | **Normalized weights** | **0.11** | **0.02** | **0.11** | **0.51** | **0.26** |

5    Table 5. Multi-criterion preference index values calculated using entropy-based weights for the Ganga River basin at Farakka site for 14 RCMs

| RCM No. | 1 | 2 | … | 7 | 8 | 9 | … | 13 | 14 |
|---|---|---|---|---|---|---|---|---|---|
| 1 | 0.00 | 0.51 | … | 0.26 | 0.00 | 0.13 | … | 0.00 | 0.39 |
| 2 | 0.49 | 0.00 | … | 0.27 | 0.02 | 0.13 | … | 0.02 | 0.39 |
| 3 | 1.00 | 0.98 | … | 0.27 | 0.02 | 0.13 | … | 0.02 | 0.49 |
| 4 | 0.74 | 0.61 | … | 0.27 | 0.02 | 0.13 | … | 0.02 | 0.49 |
| 5 | 0.74 | 0.61 | … | 0.27 | 0.02 | 0.13 | … | 0.02 | 0.49 |
| 6 | 0.24 | 0.61 | … | 0.26 | 0.02 | 0.13 | … | 0.02 | 0.49 |
| 7 | 0.74 | 0.73 | … | 0.00 | 0.12 | 0.24 | … | 0.24 | 0.74 |
| 8 | 1.00 | 0.98 | … | 0.76 | 0.00 | 1.00 | … | 1.00 | 1.00 |
| 9 | 0.87 | 0.87 | … | 0.76 | 0.00 | 0.00 | … | 0.00 | 0.89 |
| 10 | 1.00 | 1.00 | … | 0.49 | 0.49 | 0.49 | … | 0.49 | 1.00 |
| 11 | 0.76 | 0.76 | … | 0.76 | 0.00 | 0.51 | … | 0.51 | 0.76 |
| 12 | 1.00 | 0.98 | … | 1.00 | 0.24 | 1.00 | … | 1.00 | 1.00 |
| 13 | 1.00 | 0.98 | … | 0.76 | 0.00 | 1.00 | … | 0.00 | 1.00 |
| 14 | 0.61 | 0.61 | … | 0.26 | 0.00 | 0.11 | … | 0.00 | 0.00 |



Table 6. Values of $\phi+$, $\phi-$, $\phi$ and ranks of regional climate models (RCMs) for Ganga River basin at Farakka

| RCM No. | $\phi^+$ | $\phi$ | $\phi$ | Rank |
|---|---|---|---|---|
| 1 | 0.21 | 0.79 | -0.57 | 12 |
| 2 | 0.21 | 0.79 | -0.58 | 13 |
| 3 | 0.47 | 0.53 | -0.05 | 8 |
| 4 | 0.34 | 0.66 | -0.32 | 9 |
| 5 | 0.26 | 0.74 | -0.48 | 11 |
| 6 | 0.19 | 0.81 | -0.61 | 14 |
| 7 | 0.50 | 0.49 | 0.00 | 7 |
| 8 | 0.92 | 0.07 | 0.85 | **1** |
| 9 | 0.61 | 0.39 | 0.21 | 5 |
| 10 | 0.77 | 0.23 | 0.53 | **3** |
| 11 | 0.57 | 0.43 | 0.13 | 6 |
| 12 | 0.90 | 0.10 | 0.80 | **2** |
| 13 | 0.74 | 0.26 | 0.49 | 4 |
| 14 | 0.30 | 0.70 | -0.41 | 10 |

Table 7. Ranks of RCMs based on surface runoff estimated from SWAT model for Ganga basin at different
5    locations

| Locations | Rank1 | Rank2 | Rank3 |
|---|---|---|---|
| Farakka | RCA4 (CNRM-CERFACS) | RCA4 (MIROC) | RCA4 (ICHEC) |
| Kuldah Bridge | RCA4 (MPI-M) | RCA4 (MIROC) | RCA4 (CNRM-CERFACS) |
| Chopan | RCA4 (MPI-M) | RCA4 (CNRM-CERFACS) | RCA4 (MIROC) |
| Triveni | REMO2009 (MPI) | CCAM (CNRM) | CCAM (ACCESS) |
| Chatara | REMO2009 (MPI) | CCLM4 (MPI) | CCAM (BCCR) |
| Elgin Bridge | CCAM (CNRM) | CCAM (GFDL) | CCLM4 (MPI) |
| Dholpur | RegCM4 (LMDZ) | REMO2009 (MPI) | CCAM (MPI) |
| Kalanaur | RCA4 (ICHEC) | RCA4 (MIROC) | RCA4 (CNRM-CERFACS) |
| Rishikesh | RCA4 (ICHEC) | RCA4 (CNRM-CERFACS) | RCA4 (MIROC) |
| Kanpur | RCA4 (CNRM-CERFACS) | RCA4 (MIROC) | RCA4 (ICHEC) |