# Peer review of "Spatial Extent of Future Changes in the Hydrologic Cycle Components in Ganga Basin using Ranked CORDEX RCMs"

_Hydrology and Earth System Sciences, 2017_

## Referee Comment (RC1) · Anonymous Referee #1 · 27 Jun 2017

The paper presents downscaled outputs for hydrological variables in the Ganga river basin. Although the title conveys otherwise, in my opinion, the paper presents nothing more than standard studies using GCM/RCM simulations in conjunction with hydrological models to provide future hydrological projections, except that this study also ranks the climate models. In fact, instead of 'Spatial Extent of Future Changes in the Hydrologic Cycle Components in Ganga Basin using Ranked CORDEX RCMs', the paper could have simply been titled 'Downscaled hydrologic projections in Ganga Basin using Ranked CORDEX RCMs'. As a case study the paper may be of value, however, the current version of the manuscript falls seriously short on scientific contributions or providing new insights for the region. For example, I referred to three other papers

recently published in HESS on related topics as follows:

Maurer, E. P., Ficklin, D. L., and Wang, W.: Technical Note: The impact of spatial scale in bias correction of climate model output for hydrologic impact studies, Hydrol. Earth Syst. Sci., 20, 685-696, doi:10.5194/hess-20-685-2016, 2016.

Grouillet, B., Ruelland, D., Vaittinada Ayar, P., and Vrac, M.: Sensitivity analysis of runoff modeling to statistical downscaling models in the western Mediterranean, Hydrol. Earth Syst. Sci., 20, 1031-1047, doi:10.5194/hess-20-1031-2016, 2016.

Guimberteau, M., Ciais, P., Ducharne, et al.: Impacts of future deforestation and climate change on the hydrology of the Amazon Basin: a multi-model analysis with a new set of land-cover change scenarios, Hydrol. Earth Syst. Sci., 21, 1455-1475, doi:10.5194/hess-21-1455-2017, 2017.

Each of these studies present at least some new scientific insights that can guide further studies in these areas. The present study, on the other hand, simply reports statistical findings for the region under consideration instead of providing actual physical reasons that can help in process understanding or discussing justifications for the choices they make, nor does it ask or answer any new questions. Firstly, the authors should justify why only climate change may play a major role in hydrology of the basin instead of changes in landuse/land cover, particularly in this largely irrigated areas with changing cropping patterns and rapid urbanization. What role does human water withdrawal play, and how are they accounting for that in their analysis?

The authors also present scenarios till 2100. Is it sensible at all to assume that the LU/LC patterns, the vegetation patterns and the irrigation technologies will remain unaltered till 2100? If these processes have changed in the last 30 years, it is likely that they will change in the future as well. Many studies are now looking at LU/LC projections (e.g. the third paper listed above).

Further, bias correction is something that one should be cautious about. The climate

model simulations are obtained by conserving laws of physics. However, if some statistical exercise is carried out on their output to deliberately match observations, one may violate basic physics of the problem. Moreover, bias correction can map unrelated variables and distribution-based methods (such as the QQ mapping) may not detect lack of a link (refer to papers authored by D Maraun). In fact, some experts liken the process of bias correction to 'transforming elephants to mouse' (Will hydrologists learn from the world around them?: Empiricism, models, uncertainty and stationarity, U Lall, AGU Fall Meeting Abstracts 1, 01, 2010)! Therefore, the ability of the current analysis to inform practitioners and water managers at the end of the next century is questionable.

Next, why SWAT? Why not others? And why RCP4.5? Though umpteen number of studies used SWAT earlier for similar analyses (many from the authors' institute), here, the authors are attributing some of the future changes to alterations in snowmelt (paragraph 20 for example). Have the authors used any snow module in SWAT? Simply considering the low flows as snowmelt may not be accurate. What about groundwater? What is its relative contribution to low flows? If more precipitation falls as rain and less as snow (as expected from global warming), this translates into an increase of winter discharge and an increase in recharge and base flows, but the paper reports otherwise (para 15). The locations considered in this analysis (for example, Rishikesh) are not regions of permanent snowpack. Also, RCP8.5 is the most realistic scenario based on evolution of the current climate since 2005. Not clear why the authors chose RCP4.5.

The authors might want to include a plot of historical and future monthly climatology. Since the region is highly monsoon dominated, why should one expect changes in winter precipitation (say, more precipitation falling as rain instead of snow) to cause major alterations in monsoon rains and resultant runoff?

Further, I don't agree with the authors that RCMs have 'been a boon to study changing climate conditions'. Studies have shown that CORDEX RCMs show little improvement over their parent GCMs (Singh et al., Clim. Dyn., 2017) for the Indian monsoon region. Firstly, the authors' evaluation of the RCMs on simulated runoff might be contaminated

by performance of the SWAT model in differing precipitation ranges; therefore, they should be evaluated on a variable that they simulate, namely, precipitation.

More importantly, why do these three models rank better than the others? Are they permitting more physical processes (such as convection?) or are their parameterizations any different than the rest? How sensitive are they to the variables and indices considered for ranking? In general, the authors state that flows are likely to 'increase' or 'decrease' in future. It may be worthwhile to examine whether such changes are actually statistically significant as compared to the observational record. Again, examining the causes of such change would rather be a more meaningful contribution.

Finally, the performance of the CORDEX RCMs in simulating the Indian monsoon is not known to be satisfactory. If large-scale monsoon features are not well represented by these models, any 'good performance' of these models on simulated runoff may be an artefact of exercises on the data or of using calibrated SWAT. For conclusions on seasonal analysis as attempted in this paper, this needs to be re-examined.

Other comments: How are the performances of the model for different gauging locations combined? Not clear from the description provided. What drainage data has been used for SWAT? Is the calibration and validation of SWAT done with monthly data? How do the authors make inferences on extremes with monthly data? Fig. 7 and similar plots: the box plots should be explained within the figure caption. Fig. 9: since these are discrete locations, it is not correct to join them. Fig. 11: 'obs' also presented, while figure caption says 'projected'. Para 10: What are 'sudden changes'? Is 'climate change' a 'sudden' change?

---

## Referee Comment (RC2) · Anonymous Referee #2 · 12 Sep 2017

This paper presents a case study in which a hydrological model is forced with the output of CORDEX RCMs to obtain spatially-explicit information about future changes in some hydrological variables across the Ganga basin. The manuscript is quite difficult to follow and could be substantially reduced in length. There are some structural changes which could be made in order to help readers. In addition, the authors must provide better arguments for some of their decisions. For example, what are the reasons for using SWAT? Why was RCP4.5 used? Why were the RCMs evaluated on the basis of streamflow and not precipitation?

The hydrological model has been calibrated with discharge measurements at various

points on the Ganga and its tributaries. However, this river system is both highly regulated by surface water infrastructure and strongly influenced by anthropogenic activities within the basin. Clearly the assumption that such factors are stationary is false (as the authors acknowledge), which leads me to question the validity and utility of projections of surface runoff using this approach. This should be discussed in more detail. Further, the authors do not really justify their choice of SWAT. Why not a process-based, distributed model such as VIC, which has previously been used to assess hydrological change in the Ganga basin (e.g. Chawla and Mujumdar (2015), who also consider land use change)? What is the reason for choosing to rank RCMs according to surface runoff? In my opinion the performance of RCMs should be evaluated using precipitation. Do the same three models perform best if you use precipitation, not surface runoff, for evaluation purposes?

Throughout the manuscript the authors attribute some of the change in surface runoff to changes in snowmelt. While snowmelt is undoubtedly an important factor in the Himalayas, I cannot imagine the relative contribution in, say, Farakka is very high. Maybe I'm wrong, but in any case the authors must first quantify the contribution of snowmelt to surface runoff and place any discussion of the impact of changes in snowmelt in this context. Since the aim of the paper is to assess the spatial extent of future changes in hydrology it should quantify the relative contribution of various hydrological processes to streamflow at different points in the basin. This would be an interesting study. Presently, however, readers are left with little understanding about the reason for the spatial variability in surface runoff apart from the precipitation pattern.

Some more specific comments are listed below:

Page 2, line 18: "The imbalance between the water demands and water supply has been increasing of late due to change in climate" - what evidence is there for this? Is climate change more or less important than other factors such as population and economic growth? Page 2, line 34: This is the first mention of objectives. Perhaps they could be stated explicitly beforehand. Page 4, line 1: "Furthermore, the Ganga river

basin is highly governed by snowfall accumulation and snowmelt processes within the catchment" - This may be true in the Himalaya region but once the river enters the Gangetic plains I can't imagine that snowmelt contributes much to the overall flow. The following claim, that "snowmelt processes provide most of the surface water during non-monsoon season" is at best misleading and should be explained further. Is it the case that discharge at Farukka during winter months is predominantly snowmelt? What is the role of groundwater? Page 4, line 25: "The primary sources of water in the river are precipitation, base flow, and snowmelt water from the Himalayas" - again, this statement fails to recognise that the relative importance of these sources changes as the river moves downstream. Meltwater may well be an important component of discharge in Rishikesh or Haridwar, but in Patna or Farukka, say, I would imagine it's contribution would be very small. In any case the contribution of the various hydrological processes should be quantified. Page 4, line 19: This section essentially repeats what you have said in the final paragraph of the Introduction, and could be removed. Page 6, Table 1: Use consistent units for spatial resolution (e.g. SRTM @ 3 arc second, soil @ 5 arc minute, etc.) Page 6, line 5: Explain why RCP 4.5 was chosen. Page 7, Model description: Explain why SWAT was chosen. For example, why not choose a spatially distributed model for the analysis, given that the expressed aim of the paper is to assess the spatial extent of changes to the hydrologic cycle? Page 7, line 3: Why is it relevant that the model is computationally efficient? In fact these first two sentences does not fit very well and the paragraph would make more sense if it started with the third sentence. Page 7, line 10: Not sure what is meant by 'vivid methods' - perhaps 'various methods' would be more appropriate. Page 7, line 13: 'The SWAT model is used to assess the impact of land use changes...' would be more clear. Moreover it would help readers if you provide some references to back up this assertion. Page 7, line 7: Given the study aims to provide spatially explicit insight, it would be helpful if at this stage a figure was provided to show how SWAT is spatially discretised. Page 7, line 17: The Methodology does not include any description of the hydrological modelling. I see that you describe the model set-up in the section 6.1 – would it not make

more sense to put this in the methodology? Page 11, line 23: See previous comments about the relative contribution of snowmelt to baseflow. Why is baseflow attributed to snowmelt? What is the relative importance of groundwater? Page 11, line 24: It's not clear what is meant by 'intrinsic nature' - consider rewording. Page 21, Fig. 9: The x-axis represents discrete locations so the points shouldn't be joined.

References:

Chawla, I. and Mujumdar, P. P.: Isolating the impacts of land use and climate change on streamflow, Hydrol. Earth Syst. Sci., 19, 3633-3651, https://doi.org/10.5194/hess-19-3633-2015, 2015.

---

## Author Comment (AC1) · 10 Oct 2017

**Response to comments of Reviewers**

**Manuscript Number**:

**Title**: Spatial Extent of Future Changes in the Hydrologic Cycle Components in Ganga Basin using Ranked CORDEX RCMs

**Authors**: Jatin Anand, Manjula Devak, Ashvani Kumar Gosain, Rakesh Khosa and Dhanya C.T.

We would like to express our gratitude to the referees for constructive and helpful comments. We have revised the manuscript based on these comments and hope that the revised manuscript reads better and is acceptable. Our specific responses to the reviewer's comments are given below.

**Response to Comments of Reviewer #1:**

**Comment:** The authors should justify why only climate change may play a major role in hydrology of the basin instead of changes in landuse/land cover, particularly in this largely irrigated areas with changing cropping patterns and rapid urbanization. What role does human water withdrawal play, and how are they accounting for that in their analysis?

**Response:** We agree with the reviewer's comment on the combined impact (might be positive and negative) of landuse landcover (LULC), human interventions, climate change etc. in the hydrology of any basin. However, in this study, our focus was mainly on the effect of climate change on the basin hydrology, as highlighted in section 1, page 4 of the revised manuscript. The main objective of this paper is not about testing the combined effect of endogenous and exogenous changes over a catchment, but rather it is about understanding the effects of climate on hydrology, which is accomplished using a rigorously calibrated and validated hydrologic model over the basin (Anand, J., Gosain, A.K., Khosa, R., Srinivasan, R., 2017. Regional Scale Hydrologic Modeling for Prediction of Water Balance, Analysis of Trends in Streamflow and Variations in Streamflow: The case study of the Ganga River Basin. J. Hydrol. Reg. Stud.). Thus, in this study, we attempt to simulate the catchment's response to mere changes in climate, while agreeing to the reality that the actual future response will vary, based on the local human activities including LULC changes. This is acknowledged in section 7, page 26 of the revised manuscript.

Though, our focus was on climate change aspects only, in this study, proper attention has been given to the source of irrigation, thus capturing the proper hydrology of the basin in the model, however. Moreover, in SWAT model, the domestic demand calculated based on the population for each region, is also incorporated, for simulating the realistic present scenario. Having said so, we agree that there would be considerable change in the irrigation areas, changes in irrigation methods and increase in the population which in turn increase the domestic water demand, in the future. These futuristic changes are not considered in the model, however, as have been attempted by other studies carried out in this context (Oguntunde and Abiodun, 2013; Troin et al., 2015).

The effect of LULC and human withdrawals, on the hydrology will be taken up as a future study, with carefully designed experiments. This is acknowledged in section 7, page 26 of the revised manuscript.

--------------------------------------X-------------------------------------------X-----------------------

**Comment:** The authors also present scenarios till 2100. Is it sensible at all to assume that the

LU/LC patterns, the vegetation patterns and the irrigation technologies will remain un-altered till 2100? If these processes have changed in the last 30 years, it is likely that they will change in the future as well.

**Response:** As mentioned earlier that the authors very much agree with the reviewer on this part. This comment
is even applicable to the 21$^{st}$ century projections issued by IPCC through CMIP5 experiments, questioning the reliability of those projections, since those are also by assuming a few scenarios.

However, as mentioned earlier, our focus was to evaluate the spatio-temporal extent of the changes in the hydrology of the basin, due to the effect of plausible changes in the future climate only, as highlighted in section 1, page 4, line 11 in the revised manuscript.

The LULC patterns, the vegetation patterns and the irrigation technologies will certainly change in the future. Along with reiterating the fact that these were not the focus in the present study, the inclusion of all these factors in one model for a huge basin of spatial extent about $1.08 \times 10^6$ km$^2$, along with the expected spatio-temporal heterogeneity, to simulate the catchment's response in future, is a herculean task which will carry great amount of uncertainty. We anticipate that instead of including all these factors together, the individual effect of factors
might be more helpful to appreciate the possible adverse effect of these individual factors.

In addition, since LU/LC is an important input parameter, we have tried to capture the present LULC condition in the current model at a very fine level. All the landuse pattern, along with the cropping pattern and source of irrigation are included. Further, the landuse layers were kept constant, to analyse the possible impact of climate change in the spatio-temporal changes in the hydrological variables. The individual and combined effect of these
factors will be taken up as a future study, on the same basin.

---------------------------------------X-------------------------------------------X-----------------------

**Comment:** Next, why SWAT? Why not others? And why RCP4.5?

**Response:** As rightly mentioned by the reviewer, a wide range of hydrological models are available (e.g., SWAT, VIC, TOPMODEL etc.). However, SWAT is deployed in the present study, due to its widely-known versatility. Arnold et al. (1999) in his study used SWAT model to simulate the entire U.S.A. for river discharges at around
6000 gauging stations. Gosain et al., (2006) modeled twelve large river catchments in India with the purpose of quantifying the climate change impact on hydrology. Faramarzi et al., (2013) used the African model to study the impact of climate change in Africa. Abbaspour et al., (2015) simulated hydrology and water quality for entire Europe. Also. the larger capacity of ArcGIS now allows building finer resolution large-scale models, which could be calibrated using powerful parallel processing (Rouholahnejad et al., 2012). It includes a snow module,
permitting the delimitation of up to ten elevation bands with associated precipitation and temperature lapse rates (Fontaine et al., 2002; Rahman et al., 2013). Also, SWAT allows users to define simulation of irrigation water on cropland on the basis of five alternative sources: stream reach, reservoir, shallow aquifer, deep aquifer, or a water body source external to the watershed (Gassman et al., 2007). In addition to it, SWAT provide the users Reservoir operation and inclusion of domestic water demands, which other models lack. Furthermore, SWAT allows the
users to limit the size of catchment based on the gauge and reservoir sites, which the other grid based models lack and is an important factor considering the size of the basin. The wide range of SWAT applications, underscores that the model is a very flexible and robust tool that can be used to simulate a variety of watershed problems. Moreover, the process of developing SWAT model for a given watershed has also been significantly facilitated by the development of GIS-based interfaces, which provide a direct means of translating digital topographic, land use, and soil data into model inputs. Also, it is an open source code which ensures the improvement in the simulation accuracy of key processes.

**Manuscript is revised by adding the following lines. Vide Pg. 7, lines 11-18 of the revised manuscript.**

*"SWAT has inbuilt snow module to capture and simulate snowmelt hydrology, permitting the delimitation of up to ten elevation bands with associated precipitation and temperature lapse rates (Fontaine et al., 2002; Rahman et al., 2013). In addition, SWAT allows users to add lapse rate for both precipitation (PLAPS) and temperature (TLAPS) which is very essential in the study region considered. In this study, sub-basin has been divided into ten elevation bands. Furthermore, in SWAT the date and net volume of irrigation water can be set according to the actual condition along with the irrigation water source (surface rivers, reservoirs or groundwater, etc). Also, the domestic, ecological and industrial water in the model can be added in the model as "consumptive use water"."*

In the IPCC 4th Assessment Report, the emission scenarios were fabricated in view of the storylines that were gathered into an all the more financially concerned advancement or a more ecological and sustainable advancement, and into a more globalized world or an all the more regionally developing world. While, In the IPCC 5th Assessment Report, the scenarios are based on total anthropogenic radiative forcings towards the end of the 21st century. Economic models can take diverse ways to achieve distinctive radiative forcings that are equivalent to distinctive concentration paths of the greenhouse gases, the so-called Representative Concentration Pathway scenarios (RCPs). The different scenarios are marked as: RCP 8.5, RCP 4.5 and RCP 2.6, which correspond to radiative forcings of 8.5 $Wm^{-2}$, 4.5 $Wm^{-2}$, and 2.6 $Wm^{-2}$, respectively. The first RCP is the most skeptical and results in a worldwide warming towards the end of the 21st century of about 4˚C, whereas the last RCP is the most optimistic and corresponds to a global warming of about 1˚C. Then again, RCP 4.5 can be reasoned as neither extreme (RCP 8.5), nor mild (RCP 2.5); but the moderate degree scenario, which can be helpful in capturing the average response of the catchment. The RCP 4.5 resembles around to B1 scenario in IPCC 4th Assessment Report, the radiative forcing increases almost linearly up to about the year 2060 and then slows down the increase rate until the end of the century where it stabilizes. Moreover, the forcing pathway of the RCP4.5 scenario is similar to a several climate policy situations and numerous low-emissions reference conditions, such as the SRES B1 scenario. In this work, the results are shown for RCP 4.5 scenario. However, the validated model may be run for other scenarios also, and can be included.

**Manuscript is revised by adding the following lines. Vide Pg. 6, lines 8-26 of the revised manuscript.**

*"In the IPCC 4th Assessment Report, the emission scenarios were fabricated in view of the storylines that were gathered into an all the more financially concerned advancement or a more ecological and sustainable advancement, and into a more globalized world or an all the more regionally developing world. While, In the IPCC 5th Assessment Report, the scenarios are based on total anthropogenic radiative forcings towards the end of the 21st century. Economic models can take diverse ways to achieve distinctive radiative forcings that are*

*equivalent to distinctive concentration paths of the greenhouse gases, the so-called Representative Concentration*
*Pathway scenarios (RCPs). The different scenarios are marked as: RCP 8.5, RCP 4.5 and RCP 2.6, which*
*correspond to radiative forcings of 8.5 Wm$^{-2}$, 4.5 Wm$^{-2}$, and 2.6 Wm$^{-2}$, respectively. The first RCP is the most*
*skeptical and results in a worldwide warming towards the end of the 21st century of about 4˚C, whereas the last*

*RCP is the most optimistic and corresponds to a global warming of about 1˚C. Then again, RCP 4.5 can be*
*reasoned as neither extreme (RCP 8.5), nor mild (RCP 2.5); but the moderate degree scenario, which can be*
*helpful in capturing the average response of the catchment. The RCP 4.5 resembles around to B1 scenario in*
*IPCC 4th Assessment Report, the radiative forcing increases almost linearly up to about the year 2060 and then*
*slows down the increase rate until the end of the century where it stabilizes. Apparently, RCP 4.5 is a stabilization*

*scenario in which total radiative forcing is stabilized just after 2100, without exceeding the long-run radiative*
*forcing target level. Moreover, the forcing pathway of the RCP4.5 scenario is similar to a several climate policy*
*situations and numerous low-emissions reference conditions, such as the SRES B1 scenario (Wise et al., 2009).*
*Thus, in this study, RCP 4.5 scenario is adopted to study the impact of climate change on the different hydrologic*
*components and the results are shown for the same."*

[Figure]

**Comment:** Have the authors used any snow module in SWAT?

**Response:** SWAT has inbuilt snow module to capture and simulate snowmelt hydrology. SWAT model allows
the users to split the sub-basin into elevation bands. In this study, we have divided the sub-basin into ten elevation
bands. In addition, SWAT also allows users to add lapse rate for both precipitation (PLAPS) and temperature
(TLAPS) which is very essential in the study region considered.

In the present work, the emphasis has been given to the snowmelt hydrology of the basin, which was missing in
most of the earlier studies (Chawla and Mujumdar, 2015; Kothyari et al., 1997). We have used PLAPS and TLAPS
based on rigorous calibration and validation to capture the snowmelt hydrology of high altitude areas, and
comparison of observed and simulated runoffs do provide sufficient evidence that the model was able to capture
the inherent snowmelt hydrology of the region quite well. Furthermore, segregation of the percentage change in contribution of snow melt runoff from the total streamflow is done in the present work which was missing in the
earlier studies (Chawla and Mujumdar, 2015; Kothyari et al., 1997) (Fig 1 in the supplementary material).

**Manuscript is revised by adding the following lines. Vide Pg. 7, lines 11-18 of the revised manuscript.**
*"SWAT has inbuilt snow module to capture and simulate snowmelt hydrology, permitting the delimitation of up*

*to ten elevation bands with associated precipitation and temperature lapse rates (Fontaine et al., 2002; Rahman*
*et al., 2013). In addition, SWAT allows users to add lapse rate for both precipitation (PLAPS) and temperature*
*(TLAPS) which is very essential in the study region considered. In this study, sub-basin has been divided into ten*
*elevation bands. Furthermore, in SWAT the date and net volume of irrigation water can be set according to the*
*actual condition along with the irrigation water source (surface rivers, reservoirs or groundwater, etc). Also, the*

*domestic, ecological and industrial water in the model can be added in the model as "consumptive use water"."*

----------------------------------------X----------------------------------------X----------------------

**Comment:** What about groundwater? What is its relative contribution to low flows?

**Response:** Calculations has been done to find out the contribution of groundwater to stream lows. Results pertaining to the quantitative assessment of the contribution of climate change to the groundwater and base flow are provided in Section 6.4 (Pg. 24) and Figure 12 of the HESS paper and Figure S2 and S3 of supplementary material.

It is said in the manuscript that the base flow in the future would change in the range from +5000 to –1000 mm (+ 100% to -500%) (Section 6.4, Pg. 24, line 8).

Also, results pertaining to the quantitative assessment of the different hydrologic components for different tributaries of Ganga basin has been added in Section 6.4 (Pg. 26)

**Manuscript is revised by adding the following lines. Vide Pg. 26, lines 4-11 of the revised manuscript.**

"*Apparently, in order to tackle the water management issues proficiently, it is important to quantify and analyze the different hydrologic components within the basin for different system. Water balance is an important component in SWAT model as it involves all the processes happening within the watershed area (Fletcher et al., 2013; Shawul et al., 2013). The analysis of the results projected by RCA4 (CNRM-CERFACS) for the different systems of Ganga river basin shows that the water yield and evapotranspiration have major shares in the water balance (Figure 13). Further examination of results reveals that the Upper Ganga River System (Upstream of Allahabad) largely contribute to the water yield in the catchment while Chambal and Sindh river systems have lowest contribution in the total water yield (Figure 13).*"

[Figure]

Figure 13. Annual water balance components for major tributaries of the Ganga river basin projected by RCA4 (CNRM-CERFACS)

----------------------------------------X----------------------------------------X-----------------------

**Comment:** If more precipitation falls as rain and less as snow (as expected from global warming), this translates into an increase of winter discharge and an increase in recharge and base flows, but the paper reports otherwise (para 15).

**Response:** As mentioned by the reviewer that there could be an increase in the winter discharge. Please note that this **is in accordance with our finding** (Figure 7, Pg. 19 in main manuscript and Figure 1 in Supplementary material). Recharge is found to be increasing, in the northern part of the basin. Only the central part of the basin and highly irrigated areas of north-western parts have shown major reduction in the recharge and base flow (Figure 12, Pg. 24 in main manuscript and Figure 2 in Supplementary material). Water resources are synchronous with rainfall. Under natural conditions, long-term trends of groundwater recharge, is a function of total precipitation, temperature and evapotranspiration.  A combination of reduced precipitation and increased temperature can have an adverse impact on recharge within the entire basin, whereas a combination of increased precipitation and decreased temperature leads to an increase in recharge. Variations in temperature can alter evapotranspiration rate, which indirectly affects the amount of groundwater recharge. Thus, temperature change impacts are more important in areas where evapotranspiration is an important part of the hydrologic cycle. The analysis of precipitation in the Ganga River Basin showed a decreasing trend in Yamuna basin, central portion of Ganga, Chambal and Sindh basin. However, analysis of temperature time series revealed that the temperature exhibited increasing trends during the study period. The results suggest that possible future climatic conditions would have a negative impact on recharge within north-western parts of Ganga River Basin and part of central basin. not in the south-western part, where a combination of reduced precipitation and increased temperature has negative impact on recharge and evapotranspiration increases. We tried to highlight that the central part of the basin has undergone major loss in recharge and baseflow, as the majority of the population is resided in this area and is the main consumer of the maximum available water and any change in the availability of water is going to change the dynamics of the hydrology in this area in a very severe way.

----------------------------------------X----------------------------------------X-----------------------

**Comment:** The locations considered in this analysis (for example, Rishikesh) are not regions of permanent snowpack.

**Response:** As rightly pointed out by the reviewer, Rishikesh might not be the snowpack region. However, its hydrology is highly affected by the upstream areas, which are dominant with snow and glacier and Ganga itself originates from this area. In the present study, Rishikesh station was chosen, because of the availability of historical data at this location.

**Manuscript is revised by adding the following lines. Vide Pg. 22, lines 7-9 of the revised manuscript.**

*"Snow accumulation and snowmelt processes are dominant process in the upstream portion of Rishikesh, where*
*snowmelt driven flow provides most of the surface water during non-monsoon season."*

------------------------------------------X-------------------------------------------X-----------------------

**Comment:** Not clear why the authors chose RCP4.5.

**Response:** In the IPCC 4th Assessment Report, the emission scenarios were fabricated in view of the storylines that were gathered into an all the more financially concerned advancement or a more ecological and sustainable advancement, and into a more globalized world or an all the more regionally developing world. While, In the IPCC 5th Assessment Report, the scenarios are based on total anthropogenic radiative forcings towards the end of the 21st century. Economic models can take diverse ways to achieve distinctive radiative forcings that are
equivalent to distinctive concentration paths of the greenhouse gases, the so-called Representative Concentration Pathway scenarios (RCPs). The different scenarios are marked as: RCP 8.5, RCP 4.5 and RCP 2.6, which correspond to radiative forcings of 8.5 $Wm^{-2}$, 4.5 $Wm^{-2}$, and 2.6 $Wm^{-2}$, respectively. The first RCP is the most skeptical and results in a worldwide warming towards the end of the 21st century of about 4˚C, whereas the last RCP is the most optimistic and corresponds to a global warming of about 1˚C. Then again, RCP 4.5 can be
reasoned as neither extreme (RCP 8.5), nor mild (RCP 2.5); but the moderate degree scenario, which can be helpful in capturing the average response of the catchment. The RCP 4.5 resembles around to B1 scenario in IPCC 4th Assessment Report, the radiative forcing increases almost linearly up to about the year 2060 and then slows down the increase rate until the end of the century where it stabilizes. Moreover, the forcing pathway of the RCP4.5 scenario is similar to a several climate policy situations and numerous low-emissions reference
conditions, such as the SRES B1 scenario. In this work, the results are shown for RCP 4.5 scenario. However, the validated model may be run for other scenarios also, and can be included.

**Manuscript is revised by adding the following lines. Vide Pg. 6, lines 8-26 of the revised manuscript.**
*"In the IPCC 4th Assessment Report, the emission scenarios were fabricated in view of the storylines that were gathered into an all the more financially concerned advancement or a more ecological and sustainable advancement, and into a more globalized world or an all the more regionally developing world. While, In the IPCC 5th Assessment Report, the scenarios are based on total anthropogenic radiative forcings towards the end of the 21st century. Economic models can take diverse ways to achieve distinctive radiative forcings that are*
*equivalent to distinctive concentration paths of the greenhouse gases, the so-called Representative Concentration Pathway scenarios (RCPs). The different scenarios are marked as: RCP 8.5, RCP 4.5 and RCP 2.6, which correspond to radiative forcings of 8.5 $Wm^{-2}$, 4.5 $Wm^{-2}$, and 2.6 $Wm^{-2}$, respectively. The first RCP is the most skeptical and results in a worldwide warming towards the end of the 21st century of about 4˚C, whereas the last RCP is the most optimistic and corresponds to a global warming of about 1˚C. Then again, RCP 4.5 can be*
*reasoned as neither extreme (RCP 8.5), nor mild (RCP 2.5); but the moderate degree scenario, which can be*

*helpful in capturing the average response of the catchment. The RCP 4.5 resembles around to B1 scenario in IPCC 4th Assessment Report, the radiative forcing increases almost linearly up to about the year 2060 and then slows down the increase rate until the end of the century where it stabilizes. Apparently, RCP 4.5 is a stabilization scenario in which total radiative forcing is stabilized just after 2100, without exceeding the long-run radiative*

*forcing target level. Moreover, the forcing pathway of the RCP4.5 scenario is similar to a several climate policy situations and numerous low-emissions reference conditions, such as the SRES B1 scenario (Wise et al., 2009). Thus, in this study, RCP 4.5 scenario is adopted to study the impact of climate change on the different hydrologic components and the results are shown for the same."*

[Figure]

**Comment:** Firstly, the authors' evaluation of the RCMs on simulated runoff might be contaminated by performance of the SWAT model in differing precipitation ranges; therefore, they should be evaluated on a variable that they simulate, namely, precipitation.

**Response:** We beg to disagree with the reviewer. Please note that SWAT model was rigorously calibrated and validated for the entire basin, using the observed precipitation data from Indian Meteorological Department (IMD) data, as explained in section 6.1, Pg. 12. SWAT model developed for Ganga river basin is calibrated manually by following a systematic approach (Arnold et al., 2012). At first, upstream gauge is calibrated and the parameters corresponding to that gauge site are kept constant while the parameters of the sub-basins between upstream gauge and the next downstream gauge site is calibrated. The same approach was repeated for the next downstream stations till we reach the outlet of the catchment (Farakka). Once the model has been established competent enough to mimic the hydrology of the entire basin, then it is simulated using the outputs from different RCMs, using same set of calibrated parameters. Here, we tried to establish which RCM performed better in mimicking the hydrology of the basin (surface runoff in this case – the only observed variable). The comparison with respect to observed runoff, however, indirectly take into account the precipitation characteristics also, since all the other parameters used in the model are same.

**Manuscript is revised by adding the following lines. Vide Pg. 12, lines 7-13 of the revised manuscript.**

*"SWAT model developed for Ganga river basin is calibrated manually by following a systematic approach (Arnold et al., 2012). At first, upstream gauge is calibrated and the parameters corresponding to that gauge site are kept constant while the parameters of the sub-basins between upstream gauge and the next downstream gauge site is calibrated. The same approach was repeated for the next downstream stations till we reach the outlet of the catchment (Farakka). Once the model has been established competent enough to mimic the hydrology of the*

*entire basin, then it is simulated using the outputs from different RCMs, using same set of calibrated parameters."*
**Manuscript is revised by adding the following lines. Vide Pg. 16, lines 6-9 of the revised manuscript.**

*"The ranking of RCMs is done to establish, which RCM performed better in mimicking the hydrology of the basin (surface runoff in this case – the only observed variable). The comparison with respect to observed runoff,*

*however, indirectly take into account the precipitation characteristics also, since all the other parameters used in the model are same."*

----------------------------------------X----------------------------------------X-----------------------

**Comment:** More importantly, why do these three models rank better than the others? Are they permitting more physical processes (such as convection?) or are their parameterizations
any different than the rest?

**Response:** Regional climate modeling was developed based on a dynamic formulation using the initial and time-dependent lateral boundary conditions of GCMs to deliver higher resolution outputs. Due to their dedicated physics adapted to this resolution RCMs can improve some important aspects of the physical processes governing regional climate variability in precipitation and temperature, especially over watersheds located in complex topography (Troin et al., 2014 and Diffenbaugh et al., 2005). So, in a nutshell we can say that the RCMs which performed comparatively better that the other RCMs must have boundary condition and physics close to the prevailing condition. However, analysis of RCM parametrizations is beyond the scope of this study.

----------------------------------------X----------------------------------------X-----------------------

**Comment:** It may be worthwhile to examine whether such changes are actually statistically significant as compared to the observational record.

**Response:** Based on the trend analysis done by the authors for the previous studies, the analysis provides a strong indication that both climate change and anthropogenic influences have altered the water balance in the Ganga basin. It is well known that climate change can alter the hydrologic systems, especially glacier and snow melt. Thus, the increase in snow and glacier melt process may probably can cause an increase in the annual surface runoff. As a result, an increasing trend in surface runoff is observed (Figure Below). Whereas, rapid increase of settlement area, increased irrigation etc. in downstream areas and non-perennial regions have shown decreasing trend (Figure Below).

[Figure]

Figure. Trend Analysis for Various Stream Gauge Stations for Ganga river basin.

---------------------------------------X---------------------------------------X----------------------

**Comment:** Finally, the performance of the CORDEX RCMs in simulating the Indian monsoon is not known to be satisfactory.

**Response:** The NSE ranges from −∞ to 1 and measures how well the simulated versus observed data match the
1:1 line (Moriasi et al., 2007). Also, NSE is highly sensitive to the peaks (in our case monsoon flow). The analysis of the NSE values for top three ranked RCMs is greater than 0.80 (Pg. 15, Table 4), which falls in **excellent** range and shows that the RCMs are able to capture the high flows quite well in our case while, RCA4 (ICHEC) has NSE of 0.93. And except few RCMs all the others have NSE values greater than 0.7. So, we can say that performance of CORDEX RCMs in simulating Indian Monsoon is good.
A comparison has been made to average annual monthly surface runoff generated by ranked RCMs and observed surface runoff. Since, the time series data can't be given, hence average annual monthly surface runoff is plotted to showcase the efficacy of the RCMs in simulating Indian Monsoon. The results show that the RCMs has been able to simulate monsoon season appreciably.

[Figure]

Figure. Comparison of annual Monthly Observed and Simulated Discharge (1990-2004)

----------------------------------------X--------------------------------------------X-----------------------

**Comment:** What drainage data has been used for SWAT?

**Response:** In this study, surface runoff data used for the calibration and comparison purpose is provided by Central Water Commission (CWC), Ministry of Water Resources, India. This is explained in Pg. 5 line 11 – 13**.**

----------------------------------------X--------------------------------------------X-----------------------

**Comment:** Is the calibration and validation of SWAT done with monthly data? How do the authors make inferences on extremes with monthly data?

**Response:** The calibration and validation of SWAT model was done using monthly data. SWAT allows the users to run the model on daily, monthly and yearly basis. Once the model was calibrated and validated on the monthly
basis, it can be used to simulate daily flows for the same set of parameters. The simulated daily flows can be used to make analyse extremes.

----------------------------------------X--------------------------------------------X-----------------------

**References**

Abbaspour, K. C., Rouholahnejad, E., Vaghefi, S., Srinivasan, R., Yang, H. and Kløve, B.: A continental-scale hydrology and water quality model for Europe: Calibration and uncertainty of a high-resolution large-scale SWAT model, J. Hydrol., 524, 733–752, doi:10.1016/j.jhydrol.2015.03.027, 2015.

Arnold, J. G., Moriasi, D. N., Gassman, P. W., Abbaspour, K. C., White, M. J., Srinivasan, R., Santhi, C., Harmel,

R. D., Griensven, A. Van, VanLiew, M. W., Kannan, N. and Jha, M. K.: Swat: Model Use, Calibration, and Validation, Am. Soc. Agric. Biol. Eng., 55(4), 1491–1508, 2012.

Chawla, I. and Mujumdar, P. P.: Isolating the impacts of land use and climate change on streamflow, Hydrol. Earth Syst. Sci., 19(8), 3633–3651, doi:10.5194/hess-19-3633-2015, 2015.

Faramarzi, M., Abbaspour, K. C., Ashraf Vaghefi, S., Farzaneh, M. R., Zehnder, A. J. B., Srinivasan, R. and Yang, H.: Modeling impacts of climate change on freshwater availability in Africa, J. Hydrol., 480, 85–101, doi:10.1016/j.jhydrol.2012.12.016, 2013.

Fontaine, T. A., Cruickshank, T. S., Arnold, J. G. and Hotchkiss, R. H.: Development of a snowfall–snowmelt routine for mountainous terrain for the soil water assessment tool (SWAT), J. Hydrol., 262(1–4), 209–223,
doi:10.1016/S0022-1694(02)00029-X, 2002.

Gassman, P. W., Reyes, M. R., Green, C. H. and Arnold, J. G.: The soil and water assessment tool: Historical development, applications, and future research directions, Trans. Asabe, 50(4), 1211–1250, doi:10.1.1.88.6554, 2007.

Gosain, A. K., Rao, S. and Basuray, D.: Climate change impact assessment on hydrology of Indian river basins,
Current, 90(3), 346–353 [online] Available from: http://www.ias.ac.in/currsci/feb102006/346.pdf, 2006.

Kothyari, U. C., Singh, V. P. and Aravamuthan, V.: An Investigation of Changes in Rainfall and Temperature Regimes of the Ganga Basin in India, 1997.

Oguntunde, P. G. and Abiodun, B. J.: The impact of climate change on the Niger River Basin hydroclimatology, West Africa, Clim. Dyn., 40(1–2), 81–94, doi:10.1007/s00382-012-1498-6, 2013.

Rahman, K., Maringanti, C., Beniston, M., Widmer, F., Abbaspour, K. and Lehmann, A.: Streamflow Modeling in a Highly Managed Mountainous Glacier Watershed Using SWAT: The Upper Rhone River Watershed Case in Switzerland, Water Resour. Manag., 27(2), 323–339, doi:10.1007/s11269-012-0188-9, 2013.

Rouholahnejad, E., Abbaspour, K. C., Vejdani, M., Srinivasan, R., Schulin, R. and Lehmann, A.: A parallelization framework for calibration of hydrological models, Environ. Model. Softw., 31, 28–36,
doi:10.1016/j.envsoft.2011.12.001, 2012.

Troin, M., Velázquez, J. A., Caya, D. and Brissette, F.: Comparing statistical post-processing of regional and global climate scenarios for hydrological impacts assessment: A case study of two Canadian catchments, J. Hydrol., 520, 268–288, doi:10.1016/j.jhydrol.2014.11.047, 2015.

Wise, M., Calvin, K., Thomson, A., Clarke, L., Bond-Lamberty, B., Sands, R., Smith, S. J., Janetos, A. and
Edmonds, J.: Implications of limiting $CO_2$ concentrations for land use and energy., Science, 324(5931), 1183–6, doi:10.1126/science.1168475, 2009.

---

## Author Comment (AC2) · 10 Oct 2017

**Response to comments of Reviewers**

**Manuscript doi:** 10.5194/hess-2017-189, 2017

**Title**: Spatial Extent of Future Changes in the Hydrologic Cycle Components in Ganga Basin using Ranked CORDEX RCMs

**Authors**: Jatin Anand, Manjula Devak, Ashvani Kumar Gosain, Rakesh Khosa and Dhanya C.T.

We would like to express our gratitude to the referees for constructive and helpful comments. We have revised the manuscript based on these comments and hope that the revised manuscript reads better and is acceptable. Our specific responses to the reviewer's comments are given below.

**Response to Comments of Reviewer #2**

**Comment:** For example, what are the reasons for using SWAT?

**Response:** As rightly mentioned by the reviewer, a wide range of hydrological models are available (e.g., SWAT, VIC, TOPMODEL etc.). However, SWAT is deployed in the present study, due to its widely-known versatility. Arnold et al. (1999) in his study used SWAT model to simulate the entire U.S.A. for river discharges at around 6000 gauging stations. Gosain et al., (2006) modeled twelve large river catchments in India with the purpose of quantifying the climate change impact on hydrology. Faramarzi et al., (2013) used the African model to study the impact of climate change in Africa. Abbaspour et al., (2015) simulated hydrology and water quality for entire Europe. Also. the larger capacity of ArcGIS now allows building finer resolution large-scale models, which could be calibrated using powerful parallel processing (Rouholahnejad et al., 2012). It includes a snow module, permitting the delimitation of up to ten elevation bands with associated precipitation and temperature lapse rates (Fontaine et al., 2002; Rahman et al., 2013). Also, SWAT allows users to define simulation of irrigation water on cropland on the basis of five alternative sources: stream reach, reservoir, shallow aquifer, deep aquifer, or a water body source external to the watershed (Gassman et al., 2007). In addition to it, SWAT provide the users Reservoir operation and inclusion of domestic water demands, which other models lack. Furthermore, SWAT allows the users to limit the size of catchment based on the gauge and reservoir sites, which the other grid based models lack and is an important factor considering the size of the basin. The wide range of SWAT applications, underscores that the model is a very flexible and robust tool that can be used to simulate a variety of watershed problems. Moreover, the process of developing SWAT model for a given watershed has also been significantly facilitated by the development of GIS-based interfaces, which provide a direct means of translating digital topographic, land use, and soil data into model inputs. Also, it is an open source code which ensures the improvement in the simulation accuracy of key processes.

**Manuscript is revised by adding the following lines. Vide Pg. 7, lines 11-18 of the revised manuscript.**

"*SWAT has inbuilt snow module to capture and simulate snowmelt hydrology, permitting the delimitation of up to ten elevation bands with associated precipitation and temperature lapse rates (Fontaine et al., 2002; Rahman et al., 2013). In addition, SWAT allows users to add lapse rate for both precipitation (PLAPS) and temperature (TLAPS) which is very essential in the study region considered. In this study, sub-basin has been divided into ten*

*elevation bands. Furthermore, in SWAT the date and net volume of irrigation water can be set according to the actual condition along with the irrigation water source (surface rivers, reservoirs or groundwater, etc). Also, the domestic, ecological and industrial water in the model can be added in the model as "consumptive use water".*

----------------------------------------X-------------------------------------------X-----------------------

**Comment:** Why was RCP4.5 used?

10 **Response:** In the IPCC 4th Assessment Report, the emission scenarios were fabricated in view of the storylines that were gathered into an all the more financially concerned advancement or a more ecological and sustainable advancement, and into a more globalized world or an all the more regionally developing world. While, In the IPCC 5th Assessment Report, the scenarios are based on total anthropogenic radiative forcings towards the end of the 21st century. Economic models can take diverse ways to achieve distinctive radiative forcings that are

15 equivalent to distinctive concentration paths of the greenhouse gases, the so-called Representative Concentration Pathway scenarios (RCPs). The different scenarios are marked as: RCP 8.5, RCP 4.5 and RCP 2.6, which correspond to radiative forcings of 8.5 $Wm^{-2}$, 4.5 $Wm^{-2}$, and 2.6 $Wm^{-2}$, respectively. The first RCP is the most skeptical and results in a worldwide warming towards the end of the 21st century of about 4˚C, whereas the last RCP is the most optimistic and corresponds to a global warming of about 1˚C. Then again, RCP 4.5 can be

20 reasoned as neither extreme (RCP 8.5), nor mild (RCP 2.5); but the moderate degree scenario, which can be helpful in capturing the average response of the catchment. The RCP 4.5 resembles around to B1 scenario in IPCC 4th Assessment Report, the radiative forcing increases almost linearly up to about the year 2060 and then slows down the increase rate until the end of the century where it stabilizes. Moreover, the forcing pathway of the RCP4.5 scenario is similar to a several climate policy situations and numerous low-emissions reference

25 conditions, such as the SRES B1 scenario. In this work, the results are shown for RCP 4.5 scenario.

**Manuscript is revised by adding the following lines. Vide Pg. 6, lines 8-26 of the revised manuscript.**
*In the IPCC 4th Assessment Report, the emission scenarios were fabricated in view of the storylines that were gathered into an all the more financially concerned advancement or a more ecological and sustainable*

30 *advancement, and into a more globalized world or an all the more regionally developing world. While, In the IPCC 5th Assessment Report, the scenarios are based on total anthropogenic radiative forcings towards the end of the 21st century. Economic models can take diverse ways to achieve distinctive radiative forcings that are equivalent to distinctive concentration paths of the greenhouse gases, the so-called Representative Concentration Pathway scenarios (RCPs). The different scenarios are marked as: RCP 8.5, RCP 4.5 and RCP 2.6, which*

35 *correspond to radiative forcings of 8.5 $Wm^{-2}$, 4.5 $Wm^{-2}$, and 2.6 $Wm^{-2}$, respectively. The first RCP is the most skeptical and results in a worldwide warming towards the end of the 21st century of about 4˚C, whereas the last RCP is the most optimistic and corresponds to a global warming of about 1˚C. Then again, RCP 4.5 can be reasoned as neither extreme (RCP 8.5), nor mild (RCP 2.5); but the moderate degree scenario, which can be helpful in capturing the average response of the catchment. The RCP 4.5 resembles around to B1 scenario in*

40 *IPCC 4th Assessment Report, the radiative forcing increases almost linearly up to about the year 2060 and then*

*slows down the increase rate until the end of the century where it stabilizes. Apparently, RCP 4.5 is a stabilization scenario in which total radiative forcing is stabilized just after 2100, without exceeding the long-run radiative forcing target level. Moreover, the forcing pathway of the RCP4.5 scenario is similar to a several climate policy situations and numerous low-emissions reference conditions, such as the SRES B1 scenario (Wise et al., 2009). Thus, in this study, RCP 4.5 scenario is adopted to study the impact of climate change on the different hydrologic components and the results are shown for the same.ˆ'*

---------------------------------------X-------------------------------------------X-----------------------

**Comment:** Why were the RCMs evaluated on the basis of streamflow and not precipitation?

**Response:** Streamflow is an important component for ensuring proper management and ensuring sustainable use of water wealth. However, just analysing precipitation and temperature does not give us the complete picture of the inherent hydrology of the basin. As, just determining the increase or decrease in the precipitation and temperature doesn't provide any information whatsoever about the streamflow being generated or for that matter how much recharge that may happen or how much evapotranspiration occurred in the watershed. While, all the stakeholders and managers and policy makers are more likely to be interested on the availability of water for that matter. Moreover, other important hydrological components such as recharge and base flow can't be figured out just by analysing the precipitation. Also, ET can be an important factor when assessing the hydrological characteristic of the basin. Hence, it was necessary to rank the RCMs based on the end product, i.e. streamflow. Here, we tried to establish which RCM performed better in mimicking the hydrology of the basin (surface runoff in this case – the only observed variable). The comparison with respect to observed runoff, however, indirectly take into account the precipitation characteristics also, since all the other parameters used in the model are same.

**Manuscript is revised by adding the following lines. Vide Pg. 12, lines 7-13 of the revised manuscript.**

*"SWAT model developed for Ganga river basin is calibrated manually by following a systematic approach (Arnold et al., 2012). At first, upstream gauge is calibrated and the parameters corresponding to that gauge site are kept constant while the parameters of the sub-basins between upstream gauge and the next downstream gauge site is calibrated. The same approach was repeated for the next downstream stations till we reach the outlet of the catchment (Farakka). Once the model has been established competent enough to mimic the hydrology of the entire basin, then it is simulated using the outputs from different RCMs, using same set of calibrated parameters."*

**Manuscript is revised by adding the following lines. Vide Pg. 16, lines 6-9 of the revised manuscript.**

*"The ranking of RCMs is done to establish, which RCM performed better in mimicking the hydrology of the basin (surface runoff in this case – the only observed variable). The comparison with respect to observed runoff, however, indirectly take into account the precipitation characteristics also, since all the other parameters used in the model are same."*

---------------------------------------X-------------------------------------------X-----------------------

**Comment:** The hydrological model has been calibrated with discharge measurements at various points on the Ganga and its tributaries. However, this river system is both highly regulated by surface water infrastructure and strongly influenced by anthropogenic activities within the basin. Clearly the assumption that such factors are stationary is false (as the authors acknowledge), which leads me to question the validity and utility of projections of surface runoff using this approach. This should be discussed in more detail.

**Response:** The authors agree with the reviewer comment that the river system is highly regulated by surface water infrastructure and strongly influenced by anthropogenic activities. Also, the authors agree that the stationarity is not going to be there in terms of how the available water is going to be utilized in the basin. However, the authors would like to add that to determine the way the available water is going to be utilised in the basin is beyond the scope of the study. For example, the irrigation efficiency at present in India hovers around 0.35-0.45, but that can't be said to be keep on continuing as farmers are getting things automated and adopting the new and advanced practices that are currently available. Moreover, the cropping pattern in this region is quite unpredictable where few decades back there was no paddy crops grown in the Punjab and Haryana (western region of the basin) part, but has become a dominant grown crop by leap and bounds and requires huge amount of water. Also, the basin has seen the farmers moving from food crops to cash crops that have considerable impact on the seasonal availability of water. Then again, there is a continuous increase in industrial demand which cannot be pre-determined as the numbers as well as source keeps on changing.

Thus, it can be safely said that the results obtained is the most optimistic one and suffice the main objective of the study i.e. to determine the available water in the wake of changing climate and remind the policy makers and stakeholders to make the management policy for sustainable management of water before the things change for the worse.

[Figure]

--------------------------------------X-------------------------------------------X-----------------------

**Comment:** Further, the authors do not really justify their choice of SWAT. Why not a process-based, distributed model such as VIC, which has previously been used to assess hydrological change in the Ganga basin (e.g. Chawla and Mujumdar (2015), who also consider land use change)?

**Response:** As rightly mentioned by the reviewer, a wide range of hydrological models are available (e.g., SWAT, VIC, TOPMODEL etc.). However, SWAT is deployed in the present study, due to its widely-known versatility. Arnold et al. (1999) in his study used SWAT model to simulate the entire U.S.A. for river discharges at around 6000 gauging stations. Gosain et al., (2006) modeled twelve large river catchments in India with the purpose of quantifying the climate change impact on hydrology. Faramarzi et al., (2013) used the African model to study the impact of climate change in Africa. Abbaspour et al., (2015) simulated hydrology and water quality for entire Europe. Also. the larger capacity of ArcGIS now allows building finer resolution large-scale models, which could be calibrated using powerful parallel processing (Rouholahnejad et al., 2012). It includes a snow module, permitting the delimitation of up to ten elevation bands with associated precipitation and temperature lapse rates (Fontaine et al., 2002; Rahman et al., 2013). Also, SWAT allows users to define simulation of irrigation water on cropland on the basis of five alternative sources: stream reach, reservoir, shallow aquifer, deep aquifer, or a water

body source external to the watershed (Gassman et al., 2007) which is not possible in the other models. In addition to it, SWAT provide the users Reservoir operation and inclusion of domestic water demands, which other models lack. Furthermore, SWAT allows the users to limit the size of catchment based on the gauge and reservoir sites, which the other grid based models lack and is an important factor considering the size of the basin. The wide range of SWAT applications, underscores that the model is a very flexible and robust tool that can be used to simulate a variety of watershed problems. Moreover, the process of developing SWAT model for a given watershed has also been significantly facilitated by the development of GIS-based interfaces, which provide a direct means of translating digital topographic, land use, and soil data into model inputs. Also, it is an open source code which ensures the improvement in the simulation accuracy of key processes.

---------------------------------------X-------------------------------------------X-----------------------

**Comment:** What is the reason for choosing to rank RCMs according to surface runoff? In my opinion the performance of RCMs should be evaluated using precipitation. Do the same three models perform best if you use precipitation, not surface runoff, for evaluation purposes?

**Response:** Once the model has been established competent enough to mimic the hydrology of the entire basin, then it is simulated using the outputs from different RCMs, using same set of calibrated parameters. Here, we tried to establish which RCM performed better in mimicking the hydrology of the basin (surface runoff in this case – the only observed variable). The comparison with respect to observed runoff, however, indirectly take into account the precipitation characteristics also, since all the other parameters used in the model are same.

**Manuscript is revised by adding the following lines. Vide Pg. 12, lines 7-13 of the revised manuscript.**

*"SWAT model developed for Ganga river basin is calibrated manually by following a systematic approach (Arnold et al., 2012). At first, upstream gauge is calibrated and the parameters corresponding to that gauge site are kept constant while the parameters of the sub-basins between upstream gauge and the next downstream gauge site is calibrated. The same approach was repeated for the next downstream stations till we reach the outlet of the catchment (Farakka). Once the model has been established competent enough to mimic the hydrology of the entire basin, then it is simulated using the outputs from different RCMs, using same set of calibrated parameters."*

**Manuscript is revised by adding the following lines. Vide Pg. 16, lines 6-9 of the revised manuscript.**

*"The ranking of RCMs is done to establish, which RCM performed better in mimicking the hydrology of the basin (surface runoff in this case – the only observed variable). The comparison with respect to observed runoff, however, indirectly take into account the precipitation characteristics also, since all the other parameters used in the model are same."*

---------------------------------------X-------------------------------------------X-----------------------

**Comment:** Throughout the manuscript the authors attribute some of the change in surface runoff to changes in snowmelt. While snowmelt is undoubtedly an important factor in the Himalayas, I cannot imagine the relative contribution in, say, Farakka is very high. Maybe I'm wrong, but in any case the authors must first quantify the contribution of snowmelt to surface runoff and place any discussion of the impact of changes in snowmelt in this context.

**Response:** The quantification of snowmelt to snowmelt to surface runoff has been done and also impact of changes in snowmelt has been quantified (Figure S1 in supplementary material). The authors agree that the relative contribution at Farakka is very small but in the context of snow dominant and areas downstream to it is very essential.

---------------------------------------X-------------------------------------------X-----------------------

**Comment:** Since the aim of the paper is to assess the spatial extent of future changes in hydrology it should quantify the relative contribution of various hydrological processes to streamflow at different points in the basin. This would be an interesting study. Presently, however, readers are left with little understanding about the reason for the spatial variability in surface runoff apart from the precipitation pattern.

**Response:** Thanks for bringing this into the notice of the authors. Requisite changes are made during the manuscript revision. Results pertaining to the quantitative assessment of the different hydrologic components for different tributaries of Ganga basin has been added in Section 6.4 (Pg. 26)
**Manuscript is revised by adding the following lines. Vide Pg. 26, lines 4-11 of the revised manuscript.**

*"Apparently, in the order to tackle the water management issues proficiently, it is important to quantify and analyze the different hydrologic components within the basin for different system. Water balance is an important component in SWAT model as it involves all the processes happening within the watershed area (Fletcher et al., 2013; Shawul et al., 2013). The analysis of the results projected by RCA4 (CNRM-CERFACS) for the different systems of Ganga river basin show that the water yield and evapotranspiration has the major share in the water balance (Figure 13). Further examination of results reveal that the Upper Ganga River System (Upstream of Allahabad) have made the largest contribution to the water yield in the catchment while Chambal and Sindh river systems have the lowest contribution in the total water yield (Figure 13)."*

[Figure]

[Figure]

Figure 13. Annual water balance components for major tributaries of the Ganga river basin projected by RCA4 (CNRM-CERFACS)

----------------------------------------X----------------------------------------X----------------------

**Comment:** Page 2, line 18: "The imbalance between the water demands and water supply has been increasing of late due to change in climate"

**Response:**

10 **Sentence is modified in the revised manuscript, as follows. Vide Pg. 2, lines 18-20 of the revised manuscript.**

*"Moreover, the imbalance between the water demands and water supply has been increasing of late due to change in climate and rapid urbanization (Alsharhan and Wood, 2003; Guo et al., 2002; Kahya and Kalaycı, 2004)."*

----------------------------------------X----------------------------------------X----------------------

15 **Comment:** Page 2, line 34: This is the first mention of objectives. Perhaps they could be stated explicitly beforehand.

----------------------------------------X----------------------------------------X----------------------

20 **Comment:** Page 4, line 1: "Furthermore, the Ganga river basin is highly governed by snowfall accumulation and snowmelt processes within the catchment" – This

**Response:**

**Sentence is modified in the revised manuscript, as follows. Vide Pg. 4, lines 2-3 of the revised manuscript.**

25 *"Furthermore, the Northern part of the basin is highly governed by snowfall accumulation and snowmelt processes within the catchment."*

----------------------------------------X------------------------------------------X----------------------

**Comment:** The following claim, that "snowmelt processes provide most of the surface water during non-monsoon season" is at best misleading and should be explained further.

**Response:**

**Sentence is modified in the revised manuscript, as follows. Vide Pg. 4, lines 3-5 of the revised manuscript.**

*"Since snowmelt processes is one of the major contributor to the surface water during non-monsoon season in this region of the basin, its monitoring is important in hydrologic simulations."*

----------------------------------------X------------------------------------------X----------------------

**Comment:** Page 4, line 25: "The primary sources of water in the river are precipitation, base flow, and snowmelt water from the Himalayas" - again, this statement fails to recognise that the relative importance of these sources changes as the river moves downstream. Meltwater may well be an important component of discharge in Rishikesh or Haridwar, but in Patna or Farukka, say, I would imagine it's contribution would be very small. In any case the contribution of the various hydrological processes should be quantified.

**Response:** Thanks for bringing this into the notice of the authors. Requisite changes are made during the manuscript revision. Results pertaining to the quantitative assessment of the different hydrologic components for different tributaries of Ganga basin has been added in Section 6.4 (Pg. 26)

**Manuscript is revised by adding the following lines. Vide Pg. 26, lines 4-11 of the revised manuscript.**

*"Apparently, in the order to tackle the water management issues proficiently, it is important to quantify and analyze the different hydrologic components within the basin for different system. Water balance is an important component in SWAT model as it involves all the processes happening within the watershed area (Fletcher et al., 2013; Shawul et al., 2013). The analysis of the results projected by RCA4 (CNRM-CERFACS) for the different systems of Ganga river basin show that the water yield and evapotranspiration has the major share in the water balance (Figure 13). Further examination of results reveal that the Upper Ganga River System (Upstream of Allahabad) have made the largest contribution to the water yield in the catchment while Chambal and Sindh river systems have the lowest contribution in the total water yield (Figure 13)".*

[Figure]

[Figure]

Figure 13. Annual water balance components for major tributaries of the Ganga river basin projected by RCA4 (CNRM-CERFACS)

5 --------------------------------------X-----------------------------------------X----------------------

**Comment:** Page 4, line 19: This section essentially repeats what you have said in the final paragraph of the Introduction, and could be removed.

**Response:** Suggested change has been made in the revised manuscript.

10 --------------------------------------X-----------------------------------------X----------------------

**Comment:** Page 6, Table 1: Use consistent units for spatial resolution (e.g. SRTM @ 3 arc second, soil @ 5 arc minute, etc.)

**Response:** Suggested change has been made in the revised manuscript.

15 --------------------------------------X-----------------------------------------X----------------------

**Comment:** Page 6, line 5: Explain why RCP 4.5 was chosen.

**Response:** In the IPCC 4th Assessment Report, the emission scenarios were fabricated in view of the storylines that were gathered into an all the more financially concerned advancement or a more ecological and sustainable

20 advancement, and into a more globalized world or an all the more regionally developing world. While, In the IPCC 5th Assessment Report, the scenarios are based on total anthropogenic radiative forcings towards the end of the 21st century. Economic models can take diverse ways to achieve distinctive radiative forcings that are equivalent to distinctive concentration paths of the greenhouse gases, the so-called Representative Concentration Pathway scenarios (RCPs). The different scenarios are marked as: RCP 8.5, RCP 4.5 and RCP 2.6, which

25 correspond to radiative forcings of 8.5 $Wm^{-2}$, 4.5 $Wm^{-2}$, and 2.6 $Wm^{-2}$, respectively. The first RCP is the most skeptical and results in a worldwide warming towards the end of the 21st century of about 4˚C, whereas the last

RCP is the most optimistic and corresponds to a global warming of about 1˚C. Then again, RCP 4.5 can be reasoned as neither extreme (RCP 8.5), nor mild (RCP 2.5); but the moderate degree scenario, which can be helpful in capturing the average response of the catchment. The RCP 4.5 resembles around to B1 scenario in IPCC 4th Assessment Report, the radiative forcing increases almost linearly up to about the year 2060 and then slows

5     down the increase rate until the end of the century where it stabilizes. Moreover, the forcing pathway of the RCP4.5 scenario is similar to a several climate policy situations and numerous low-emissions reference conditions, such as the SRES B1 scenario. In this work, the results are shown for RCP 4.5 scenario.

**Manuscript is revised by adding the following lines. Vide Pg. 6, lines 8-26 of the revised manuscript.**

10     *"In the IPCC 4th Assessment Report, the emission scenarios were fabricated in view of the storylines that were gathered into an all the more financially concerned advancement or a more ecological and sustainable advancement, and into a more globalized world or an all the more regionally developing world. While, In the IPCC 5th Assessment Report, the scenarios are based on total anthropogenic radiative forcings towards the end of the 21st century. Economic models can take diverse ways to achieve distinctive radiative forcings that are*

15     *equivalent to distinctive concentration paths of the greenhouse gases, the so-called Representative Concentration Pathway scenarios (RCPs). The different scenarios are marked as: RCP 8.5, RCP 4.5 and RCP 2.6, which correspond to radiative forcings of 8.5 $Wm^{-2}$, 4.5 $Wm^{-2}$, and 2.6 $Wm^{-2}$, respectively. The first RCP is the most skeptical and results in a worldwide warming towards the end of the 21st century of about 4˚C, whereas the last RCP is the most optimistic and corresponds to a global warming of about 1˚C. Then again, RCP 4.5 can be*

20     *reasoned as neither extreme (RCP 8.5), nor mild (RCP 2.5); but the moderate degree scenario, which can be helpful in capturing the average response of the catchment. The RCP 4.5 resembles around to B1 scenario in IPCC 4th Assessment Report, the radiative forcing increases almost linearly up to about the year 2060 and then slows down the increase rate until the end of the century where it stabilizes. Apparently, RCP 4.5 is a stabilization scenario in which total radiative forcing is stabilized just after 2100, without exceeding the long-run radiative*

25     *forcing target level. Moreover, the forcing pathway of the RCP4.5 scenario is similar to a several climate policy situations and numerous low-emissions reference conditions, such as the SRES B1 scenario (Wise et al., 2009). Thus, in this study, RCP 4.5 scenario is adopted to study the impact of climate change on the different hydrologic components and the results are shown for the same."*

30

-------------------------------------X-----------------------------------------X----------------------

**Comment:** Page 7, Model description: Explain why SWAT was chosen.

**Response:** As rightly mentioned by the reviewer, a wide range of hydrological models are available (e.g., SWAT,

35     VIC, TOPMODEL etc.). However, SWAT is deployed in the present study, due to its widely-known versatility. Arnold et al. (1999) in his study used SWAT model to simulate the entire U.S.A. for river discharges at around 6000 gauging stations. Gosain et al., (2006) modeled twelve large river catchments in India with the purpose of quantifying the climate change impact on hydrology. Faramarzi et al., (2013) used the African model to study the impact of climate change in Africa. Abbaspour et al., (2015) simulated hydrology and water quality for entire

40     Europe. Also. the larger capacity of ArcGIS now allows building finer resolution large-scale models, which could

be calibrated using powerful parallel processing (Rouholahnejad et al., 2012). It includes a snow module, permitting the delimitation of up to ten elevation bands with associated precipitation and temperature lapse rates (Fontaine et al., 2002; Rahman et al., 2013). Also, SWAT allows users to define simulation of irrigation water on cropland on the basis of five alternative sources: stream reach, reservoir, shallow aquifer, deep aquifer, or a water body source external to the watershed (Gassman et al., 2007). In addition to it, SWAT provide the users Reservoir operation and inclusion of domestic water demands, which other models lack. Furthermore, SWAT allows the users to limit the size of catchment based on the gauge and reservoir sites, which the other grid based models lack and is an important factor considering the size of the basin. The wide range of SWAT applications, underscores that the model is a very flexible and robust tool that can be used to simulate a variety of watershed problems. Moreover, the process of developing SWAT model for a given watershed has also been significantly facilitated by the development of GIS-based interfaces, which provide a direct means of translating digital topographic, land use, and soil data into model inputs. Also, it is an open source code which ensures the improvement in the simulation accuracy of key processes.

**Manuscript is revised by adding the following lines. Vide Pg. 7, lines 11-18 of the revised manuscript.**

*"SWAT has inbuilt snow module to capture and simulate snowmelt hydrology, permitting the delimitation of up to ten elevation bands with associated precipitation and temperature lapse rates (Fontaine et al., 2002; Rahman et al., 2013). In addition, SWAT allows users to add lapse rate for both precipitation (PLAPS) and temperature (TLAPS) which is very essential in the study region considered. In this study, sub-basin has been divided into ten elevation bands. Furthermore, in SWAT the date and net volume of irrigation water can be set according to the actual condition along with the irrigation water source (surface rivers, reservoirs or groundwater, etc). Also, the domestic, ecological and industrial water in the model can be added in the model as "consumptive use water"."*

---------------------------------------X---------------------------------------X-----------------------

**Comment:** Page 7, line 3: Why is it relevant that the model is computationally efficient? In fact these first two sentences does not fit very well and the paragraph would make more sense if it started with the third sentence.

**Response:** Suggested change has been made in the revised manuscript.

---------------------------------------X---------------------------------------X-----------------------

**Comment:** Page 7, line 10: Not sure what is meant by 'vivid methods' - perhaps 'various methods' would be more appropriate.

**Response:** Suggested change has been made in the revised manuscript.

**Sentence is modified in the revised manuscript, as follows. Vide Pg. 7, lines 8-10 of the revised manuscript.**

*"Model offers various methods such as, Hargreaves, Priestley-Taylor or Penman-Monteith methods for the quantification of evapotranspiration."*

---------------------------------------X---------------------------------------X-----------------------

**Comment:** Page 7, line 13: 'The SWAT model is used to assess the impact of land use changes...' would be more clear. Moreover it would help readers if you provide some references to back up this assertion.

**Response:** Suggested change has been made in the revised manuscript.

**Sentence is modified in the revised manuscript, as follows. Vide Pg. 7, lines 18-19 of the revised manuscript.**

*"The SWAT model is used to assess the impact of land use changes on water supplies and erosion in large-scale catchments (Li et al., 2016; Marhaento et al., 2017)."*

--------------------------------------X--------------------------------------X-----------------------

**Comment:** Page 11, line 23: See previous comments about the relative contribution of snowmelt to baseflow. Why is baseflow attributed to snowmelt? What is the relative importance of groundwater?

**Response:** Suggested change has been made in the revised manuscript.

**Sentence is modified in the revised manuscript, as follows. Vide Pg. 12, lines 20-22 of the revised manuscript.**

*"High values of NSE and $R^2$ indicate that the model has been able to capture high flows, while the low values of PBias show that the low flows attributed mostly from the baseflow and snowmelt has also been captured quiet well by the model (Fig. 3)."*

The authors would like to add that the stress has been given on the snowmelt part as the various calibration sites like Rishikesh, Uttarkashi and Joshipath are higly influenced by snow hydrology. And, rigorous calibration and validation was required to calibrate these sites, especially low flows, which was snowmelt in this case.

--------------------------------------X--------------------------------------X-----------------------

**Comment:** Page 11, line 24: It's not clear what is meant by 'intrinsic nature' - consider rewording.

**Sentence is modified in the revised manuscript, as follows. Vide Pg. 12, lines 22-23 of the revised manuscript.**

*"In general, SWAT model is able to capture the inherent nature of catchment in terms of surface runoff very precisely."*

--------------------------------------X--------------------------------------X-----------------------

**References**

Abbaspour, K. C., Rouholahnejad, E., Vaghefi, S., Srinivasan, R., Yang, H. and Kløve, B.: A continental-scale hydrology and water quality model for Europe: Calibration and uncertainty of a high-resolution large-scale SWAT

model, J. Hydrol., 524, 733–752, doi:10.1016/j.jhydrol.2015.03.027, 2015.

Faramarzi, M., Abbaspour, K. C., Ashraf Vaghefi, S., Farzaneh, M. R., Zehnder, A. J. B., Srinivasan, R. and Yang, H.: Modeling impacts of climate change on freshwater availability in Africa, J. Hydrol., 480, 85–101, doi:10.1016/j.jhydrol.2012.12.016, 2013.

5  Fontaine, T. A., Cruickshank, T. S., Arnold, J. G. and Hotchkiss, R. H.: Development of a snowfall–snowmelt routine for mountainous terrain for the soil water assessment tool (SWAT), J. Hydrol., 262(1–4), 209–223, doi:10.1016/S0022-1694(02)00029-X, 2002.

Gassman, P. W., Reyes, M. R., Green, C. H. and Arnold, J. G.: The soil and water assessment tool: Historical development, applications, and future research directions, Trans. Asabe, 50(4), 1211–1250, doi:10.1.1.88.6554,

10  2007.

Gosain, A. K., Rao, S. and Basuray, D.: Climate change impact assessment on hydrology of Indian river basins, Current, 90(3), 346–353 [online] Available from: http://www.ias.ac.in/currsci/feb102006/346.pdf, 2006.

Li, J., Li, G., Zhou, S. and Chen, F.: Quantifying the Effects of Land Surface Change on Annual Runoff Considering Precipitation Variability by SWAT, Water Resour. Manag., 30(3), 1071–1084, doi:10.1007/s11269-

15  015-1211-8, 2016.

Marhaento, H., Booij, M. J., Rientjes, T. H. M. and Hoekstra, A. Y.: Attribution of changes in the water balance of a tropical catchment to land use change using the SWAT model, Hydrol. Process., 31(11), 2029–2040, doi:10.1002/hyp.11167, 2017.

Rahman, K., Maringanti, C., Beniston, M., Widmer, F., Abbaspour, K. and Lehmann, A.: Streamflow Modeling

20  in a Highly Managed Mountainous Glacier Watershed Using SWAT: The Upper Rhone River Watershed Case in Switzerland, Water Resour. Manag., 27(2), 323–339, doi:10.1007/s11269-012-0188-9, 2013.

Rouholahnejad, E., Abbaspour, K. C., Vejdani, M., Srinivasan, R., Schulin, R. and Lehmann, A.: A parallelization framework for calibration of hydrological models, Environ. Model. Softw., 31, 28–36, doi:10.1016/j.envsoft.2011.12.001, 2012.

25  Wise, M., Calvin, K., Thomson, A., Clarke, L., Bond-Lamberty, B., Sands, R., Smith, S. J., Janetos, A. and Edmonds, J.: Implications of limiting $CO_2$ concentrations for land use and energy., Science, 324(5931), 1183–6, doi:10.1126/science.1168475, 2009.

---

## Author Comment (AC3) · 10 Oct 2017

Response to Comments of Reviewer # 1

Comment: Is the calibration and validation of SWAT done with monthly data? How do the authors make inferences on extremes with monthly data?

Response: The calibration and validation of SWAT model was done using monthly data. SWAT allows the users to compile the model results on daily, monthly and yearly basis. The model is always run on the daily basis, however in the present case it was the water balance of the basin that was the focus of the climate change impacts,

monthly period for calibrated and validated of the model was considered. The simulated daily flows can be used to make analyse for extremes.

---

## Author Comment (AC4) · 10 Oct 2017

Response to Comments of Reviewer #2

Comment: Why were the RCMs evaluated on the basis of streamflow and not precipitation?

Response: In our opinion, streamflow is a better indicator of overall response of the basin and thus should be used for comparing the performance of the model with respect to the observed flow, being a single point entity. Just analysing precipitation and temperature does not give us the complete picture of the inherent hydrology of the

basin. As, just determining the increase or decrease in the precipitation and temperature doesn't provide any information whatsoever about the streamflow being generated or for that matter how much recharge that may happen or how much evapotranspiration occurred in the watershed. While, all the stakeholders and managers and policy makers are more likely to be interested on the availability of water for that matter. Moreover, other important hydrological components such as recharge and base flow can't be figured out just by analysing the precipitation. Also, ET can be an important factor when assessing the hydrological characteristic of the basin. Hence, it was necessary to rank the RCMs based on the end product, i.e. streamflow. Here, we tried to establish which RCM performed better in mimicking the hydrology of the basin (surface runoff in this case – the only observed variable). The comparison with respect to observed runoff, however, indirectly take into account the precipitation characteristics also, since all the other parameters used in the model are same.